# Antibody escape by polyomavirus capsid mutation facilitates neurovirulence

Matthew D Lauver[1†], Daniel J Goetschius[2†], Colleen S Netherby-Winslow[1], Katelyn N Ayers[1], Ge Jin[1], Daniel G Haas[1], Elizabeth L Frost[1‡], Sung Hyun Cho[3], Carol M Bator[3], Stephanie M Bywaters[4,5], Neil D Christensen[4,5], Susan L Hafenstein[2,3,6*], Aron E Lukacher[1*]

[1]Department of Microbiology and Immunology, Penn State College of Medicine, Hershey, United States; [2]Department of Biochemistry and Molecular Biology, Pennsylvania State University, University Park, United States; [3]Huck Institutes of the Life Sciences, Pennsylvania State University, University Park, United States; [4]Department of Pathology, Penn State College of Medicine, Hershey, United States; [5]The Jake Gittlen Laboratories for Cancer Research, Penn State College of Medicine, Hershey, United States; [6]Department of Medicine, Penn State College of Medicine, Hershey, United States

*For correspondence:
shafenstein@psu.edu (SLH);
alukacher@pennstatehealth.psu.edu (AEL)

[†]These authors contributed equally to this work

Present address: [‡]Center for Brain Immunology and Glia, Department of Neuroscience, School of Medicine, University of Virginia, Charlottesville, United States

Competing interests: The authors declare that no competing interests exist.

**Abstract** JCPyV polyomavirus, a member of the human virome, causes progressive multifocal leukoencephalopathy (PML), an oft-fatal demyelinating brain disease in individuals receiving immunomodulatory therapies. Mutations in the major viral capsid protein, VP1, are common in JCPyV from PML patients (JCPyV-PML) but whether they confer neurovirulence or escape from virus-neutralizing antibody (nAb) in vivo is unknown. A mouse polyomavirus (MuPyV) with a sequence-equivalent JCPyV-PML VP1 mutation replicated poorly in the kidney, a major reservoir for JCPyV persistence, but retained the CNS infectivity, cell tropism, and neuropathology of the parental virus. This mutation rendered MuPyV resistant to a monoclonal Ab (mAb), whose specificity overlapped the endogenous anti-VP1 response. Using cryo-EM and a custom sub-particle refinement approach, we resolved an MuPyV:Fab complex map to 3.2 Å resolution. The structure revealed the mechanism of mAb evasion. Our findings demonstrate convergence between nAb evasion and CNS neurovirulence in vivo by a frequent JCPyV-PML VP1 mutation.

## Introduction

The humoral immune response is critical for controlling acute and persistent viral infections; evasion of the neutralizing antibody (nAb) response often underlies virus-mediated morbidity and mortality. Seasonal influenza vaccinations are necessitated by the emergence of influenza A virus subtypes with mutations in hemagglutinin and neuraminidase capsid proteins that handicap neutralization by virus-specific antibodies (*Bedford et al., 2015*; *Hensley et al., 2009*; *Petrova and Russell, 2018*). Viruses causing persistent infections also acquire mutations that evade nAbs (*Ciurea et al., 2000*; *Inuzuka et al., 2018*; *Kinchen et al., 2018*; *Salpini et al., 2015*; *Wei et al., 2003*). The human virome is composed of a sizeable number of persistent viruses whose pathogenicity is restrained by a healthy adaptive immune system (*Virgin et al., 2009*).

JC polyomavirus (JCPyV) is a prevalent member of the human virome (*Kamminga et al., 2018*; *Viscidi et al., 2011*). Immunological perturbations are a necessary antecedent for progressive multifocal leukoencephalopathy (PML), a fatal demyelinating brain disease caused by JCPyV (*Haley and Atwood, 2017*). PML was originally described as a rare complication of hematological malignancies and its incidence dramatically increased in the pre-HAART AIDS epoch (*Astrom et al., 1958*;

*Berger et al., 1987*; *Krupp et al., 1985*). PML has 're-emerged' as a complication of immunomodulatory therapies, such as natalizumab (α4 integrin humanized mAb) for treatment of relapsing-remitting multiple sclerosis (RRMS), rituximab (CD20 humanized mAb) for chronic lymphocytic leukemia and non-Hodgkin's lymphoma, and efalizumab (LFA-1 humanized mAb) for severe plaque psoriasis (*Carson et al., 2009*; *Kleinschmidt-DeMasters and Tyler, 2005*; *Kothary et al., 2011*; *Langer-Gould et al., 2005*; *Schwab et al., 2012*; *Van Assche et al., 2005*). JCPyV isolates from the brains, cerebrospinal fluid (CSF), and blood of PML patients contain unique mutations not found in virus present in patients' urine or in circulating (archetype) strains (*Van Loy et al., 2015*; *Vaz et al., 2000*). These JCPyV-PML variants contain rearrangements in the noncoding control region (NCCR) including deletions, insertions, and duplications. These rearrangements alter transcription factor binding sites and enhance viral replication in glial cells (*Gosert et al., 2010*; *Marshall et al., 2010*); thus, NCCR rearrangements have been regarded as important for CNS tropism (*Haley and Atwood, 2017*). In addition, most JCPyV-PML variants have non-synonymous mutations in VP1, the major viral capsid protein, with the most common being a leucine-to-phenylalanine substitution at residue 54 (L54F) and a serine-to-phenylalanine/tyrosine substitution at residue 268 (S268F/Y) (*Gorelik et al., 2011*). These VP1 mutations have been reported to alter viral receptor binding, resulting in the utilization of a restricted set of receptors for cellular attachment and entry, thereby altering viral tropism (*Geoghegan et al., 2017*; *Maginnis et al., 2013*). In hypomyelinated RAG$^{-/-}$ mice engrafted with human glial precursor cells (GPCs), however, infection with wild type or VP1 mutant JCPyVs resulted in similar levels of glial cell infection (*Kondo et al., 2014*). Recent evidence has also implicated VP1 mutations as nAb escape variants. PML patient sera only weakly neutralized patient-matched JCPyV VP1 variants (*Ray et al., 2015*). nAbs recognize antigenic epitopes that may overlap with receptor-binding sites; therefore, capsid mutations can affect both cellular tropism and humoral immunity (*Kinchen et al., 2018*; *Lynch et al., 2015*; *McKnight et al., 1995*; *Reh et al., 2018*). The relative impact of VP1 mutations in JCPyV on nAb recognition and tissue tropism is unknown.

The tight species-specificity of polyomaviruses obviates investigating the role of these JCPyV-PML VP1 mutations in vivo. Mouse polyomavirus (MuPyV) shares many features with JCPyV, including asymptomatic persistent infection, viral persistence in the kidney, and control by the virus-specific adaptive immune response (*Berger et al., 2017*; *Du Pasquier et al., 2004*; *Han Lee et al., 2006*; *Szomolanyi-Tsuda and Welsh, 1996*). MuPyV and JCPyV are non-enveloped, circular dsDNA viruses with capsids ~45 nm in diameter. Their 5 kb genomes encode the nonstructural T antigen proteins and the VP1 and VP2/VP3 structural proteins. Five copies of VP1 intertwine to form each capsomer subunit that also incorporates one copy of VP2/VP3 (*Hurdiss et al., 2016*). The core secondary structure formed by VP1 is the antiparallel β barrel structure commonly called the jellyroll. The β strands (BIDG and CHEF) are connected by flexible loops (BC, DE, EF, and HI) that extend outward from the surface of the capsomer and comprise the majority of the VP1 hypervariable regions. VP1 C-terminal extensions interact to link capsomers together forming the T = 7 d icosahedron. The capsomers that occupy the 12 icosahedral fivefold vertices are referred to as pentavalent capsomers, as each of them is surrounded by five neighboring capsomers. Each of the remaining sixty capsomers in the icosahedron has six neighboring subunits and is referred to as a hexavalent capsomer (*Caspar and Klug, 1962*; *Harrison, 2017*; *Hurdiss et al., 2018*). Each asymmetric unit contains six VP1 molecules that are structurally distinct because they experience different environments. The five VP1 molecules within a pentavalent capsomer are structurally identical, whereas the five VP1 molecules within each hexavalent capsomer are quasi-equivalent (*Caspar and Klug, 1962*).

The X-ray structure of MuPyV VP1 pentamers has been solved at resolutions ranging from 1.64 Å to 2.0 Å (*Buch et al., 2015*; *Stehle and Harrison, 1997*). However, crystallization of isolated pentamers may not represent the native environment of the icosahedral capsid. There are several structures of the entire icosahedral capsid including a 3.65 Å resolution X-ray map for MuPyV and a 3.4 Å resolution cryo-EM structure of BKPyV (*Hurdiss et al., 2018*; *Stehle and Harrison, 1996*). Notably, a 4.2 Å map of BKPyV interacting with single chain variable fragment (scFv) provided insight into antibody neutralization of polyomavirus (*Lindner et al., 2019*). Structural studies typically use the fragment antigen-binding (Fab) to avoid potential cross-linking of capsids by and multiple points of intrinsic flexibility of intact antibodies that interfere with cryo-EM analysis. Use of the Fab domain (or scFv) maximizes attainable resolution at the experimentally relevant antigen-binding domain.

Recent hardware advances in cryo-EM have led to major innovations in software design that overcomes limitations resulting from particle flexibility, heterogeneity, and imperfect symmetry

(*Abrishami et al., 2020*; *Ilca et al., 2015*; *McMullan et al., 2016*; *Scheres et al., 2009*). Sub-particle refinement approaches have made possible higher resolution maps of large, flexible virus capsids (*Bhella, 2019*; *Chen et al., 2018*; *Zhu et al., 2018*). Atomic resolution structures of virus–Fab complexes can elucidate mechanisms of neutralization and define conformational epitopes on the capsid, including key residues involved in recognition by the antibody (*Dong et al., 2017*; *He et al., 2020*; *Zhu et al., 2018*). Virus–Fab complex structures may predict viable escape mutations that naturally emerge under selective pressure from nAbs.

S268 of JCPyV corresponds to V296 of MuPyV VP1, with the residues mapping to the same position on the capsid (*Sunyaev et al., 2009*). We found that MuPyV carrying the V296F VP1 mutation was impaired in its ability to replicate in the kidney, but replicated in the brain equivalently to parental virus. In addition, this mutant virus was completely resistant to neutralization by a VP1 mAb (clone 8A7H5), which recognizes a VP1 region overlapping the dominant target of the endogenous antibody response (*Swimm et al., 2010*). To determine the mechanism of humoral escape, we solved the cryo-EM structures of MuPyV in the presence and absence of the Fab of the VP1 mAb 8A7H5. Using a local refinement strategy, we attained sufficient resolution and we could build the Fab variable domain de novo. This cryo-EM analysis identified unambiguous contact residues at the interface between VP1 and Fab, and indicated the likely mechanism of immune escape by the V296F variant. This high resolution description of the 8A7H5 epitope also provided plausible neutralization escape mechanisms for other JCPyV-PML mutations and several additional MuPyV variants isolated in vitro. Together, our data demonstrate that VP1 mutations in polyomaviruses concomitantly enabled evasion of the nAb response and facilitated neurovirulence by preserving viral replication in the CNS. These findings support the concept that viremia by nAb-resistant VP1 JCPyV variants is a critical early step in PML pathogenesis.

## Results

### V296F VP1 mutation retains MuPyV tropism for brain but not kidney

To model the effects of the S268F PML mutation, we generated a MuPyV mutant containing a V296F substitution in the wild type (WT) A2 strain (A2.V296F) (*Figure 1A*; *Dawe et al., 1987*; *Sunyaev et al., 2009*). A2.V296F exhibited a slight reduction compared to WT virus in a 60-hr single-cycle replication assay, but showed equivalent expression of the nonstructural Large T antigen (LT) mRNA 24-hr post-infection (hpi) in several mouse cell lines and primary mouse embryonic fibroblasts (*Figure 1—figure supplement 1*). Thus, this VP1 mutation had little impact on the ability of MuPyV to infect and replicate in vitro.

Little is known how the JCPyV S268F mutation affects JCPyV tropism in vivo. In PML patients, VP1 mutant viruses are detected in blood, CSF, and brain tissue, but not urine (*Gorelik et al., 2011*; *Reid et al., 2011*). Because the kidney is a reservoir for both JCPyV and MuPyV persistence, the absence of JCPyV VP1 mutant virus in the urine led us to ask whether the S268F virus exhibited a defect in kidney tropism. Compared to mice inoculated subcutaneously (s.c.) with parental A2, mice given A2.V296F showed significantly lower infection levels in the kidney 4 days post-infection (dpi) (*Figure 1B*). Immunocompetent mice infected with MuPyV do not develop productive kidney infection when detected by immunofluorescence or immunohistochemistry (*Drake and Lukacher, 1998*). $Stat1^{-/-}$ mice depleted of $CD8^+$ T cells develop severe systemic MuPyV infection (*Mockus et al., 2020*). We compared A2 and A2.V296F kidney infection in $CD8^+$ T cell-depleted $Stat1^{-/-}$ mice by staining kidney sections at day 7 pi for VP1. A2 virus-infected mice developed large, numerous $VP1^+$ foci in the kidney, but A2.V296F-infected mice exhibited only small, sporadic $VP1^+$ foci (*Figure 1C*). These results demonstrated that the V296F mutation impaired the ability of MuPyV to replicate in the kidney even under conditions of profound immunosuppression.

To determine whether the V296F mutation altered infection in the CNS, we infected mice intracranially (i.c.) and examined expression of LT mRNA. At four dpi following i.c. inoculation, A2.V296F showed equivalent infection in the brain to A2 (*Figure 1D*). To test if equivalent LT mRNA levels were indicative of infection in similar cell types, we examined brains 4 dpi for $VP1^+$ cells by immunofluorescence microscopy. Infection with either virus resulted in sporadic $VP1^+$ ependymal cells (vimentin$^+$) or astrocytes (GFAP$^+$) (*Figure 1E*; *Lavado and Oliver, 2011*; *Tissir et al., 2010*). Because the S268F mutation is only seen in PML patients, we next asked whether A2 would outcompete A2.

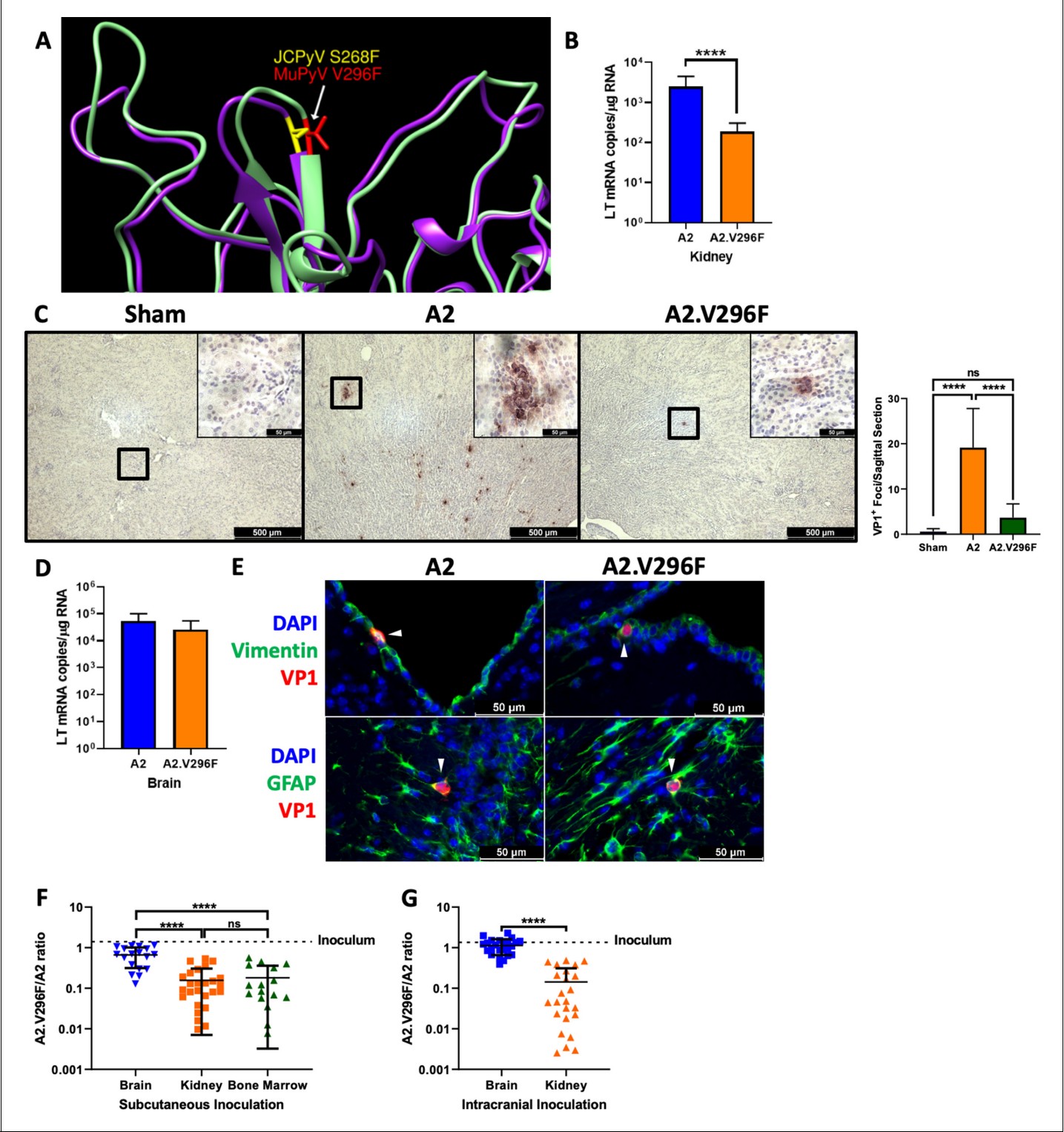

**Figure 1.** The V296F VP1 mutation in MuPyV impairs kidney, but not brain, infection. (**A**) Structural comparison of JCPyV.S268 (PDB 3NXG) and MuPyV. V296 (PDB 5CPU) VP1 residues (*Buch et al., 2015*; *Neu et al., 2010*). (**B**) A2 and A2.V296F LT mRNA levels 4 dpi in the kidneys of mice infected s.c. Data are from three independent experiments, n = 17 mice (p<0.0001). (**C**) Left: 40x images of kidneys from CD8 T cell-depleted, *Stat1*$^{-/-}$ mice 7 dpi stained for VP1. Inset is a 400x image of the region outlined in black. Right: Quantification of VP1$^+$ foci per sagittal kidney section. Data are the average of two sagittal kidney sections per mouse from three independent experiments, n = 6–11. For Sham vs. A2 p<0.0001, Sham vs. A2.V296F p=0.5801, A2 vs. A2.V296F p<0.0001. (**D**) A2 and A2.V296F LT mRNA levels four dpi in the brains of mice infected i.c. Data are from three independent experiments,
*Figure 1 continued on next page*

*Figure 1 continued*

n = 12–13 mice (p=0.1366). (E) 400x images of brains 4 dpi with A2 or A2.V296F i.c. VP1[+] cells are indicated with white arrows. Representative of three independent experiments. (F and G) Ratio of A2.V296F to A2 in various organs of mice 14 dpi with a 1:1 PFU inoculum of A2:A2.V296F s.c. (F) or i.c. (G). The dotted line indicates the ratio of A2:A2.V296F DNA in the inoculum. Data are from 2 to 3 independent experiments, n = 16–26 mice. For (F) Brain vs. Kidney p<0.0001, Brain vs. Bone Marrow p<0.0001, Kidney vs. Bone Marrow p=0.9443; for (G) p<0.0001. Data were analyzed by Mann–Whitney $U$ test (B, D, G) or one-way ANOVA (F). ****p<0.0001.

The online version of this article includes the following figure supplement(s) for figure 1:

**Figure supplement 1.** The A2.V296F VP1 mutant virus retains infectivity in vitro.
**Figure supplement 2.** Discrimination of A2 and A2.V296F DNA by PCR.

V296F in vivo. Mice received a 1:1 mixture (by PFU) of A2 and A2.V296F either i.c. or s.c.; the ratio of A2.V296F to A2 was determined at 14 dpi in various organs. To detect the relative levels of viral DNA, PCR primers were designed that amplified either a region of LT from both viruses or only the VP1 sequence of A2.V296F (*Figure 1—figure supplement 2*). In mice infected s.c. a reduced A2. V296F:A2 ratio was seen in both the kidney and bone marrow, but this ratio was significantly higher and nearly equal in the brain (*Figure 1F*). In i.c. inoculated mice A2.V296F infected the brain 1:1 with A2, but in the kidney was significantly outcompeted by A2, resulting in >1:100 A2.V296F:A2 ratio in the kidneys of some mice (*Figure 1G*). These data indicated that compared to WT VP1, the V296F mutation caused decreased infection in the kidney, but equivalent infection in the brain.

We next asked whether CNS infection with A2.V296F caused similar encephalopathy as the A2 virus. Infection with either virus induced pronounced hydrocephalus of the lateral ventricles at 30 dpi (*Figure 2A*) with multiple foci of ablated ependyma and dysplastic changes in the choroid plexus (*Figure 2B*). Sham-infected mice had single layer of vimentin[+] cells adjacent to the ventricles, consistent with vimentin expression in the region being largely restricted to ependymal cells (*Figure 2C*; *Tissir et al., 2010*). Brains of mice infected with either virus had an expansion of the vimentin[+] region abutting the ventricles, indicating damage to and/or disruption of the ependymal lining as a result of infection (*Figure 2C and D*). The vimentin[+] cells in this region also had increased GFAP expression, possibly representing subventricular zone neural precursors responding to the disruption of the ependymal layer (*Figure 2C*; *Chojnacki et al., 2009*). The ependyma and periventricular region had aggregates of CD3[+] T cells and were diffusely infiltrated by Iba1[+] cells (macrophages or microglia) in both A2- and A2.V296F-infected mice (*Figure 2E*). Together, these data demonstrated the V296F mutant virus retained the encephalogenic properties of the parental virus.

## V296F confers resistance to a neutralizing VP1 antibody

Because this PML-like VP1 mutation in MuPyV was indistinguishable from the parental virus in CNS tropism and pathology, we explored the possibility that V296F allowed evasion of anti-polyomavirus humoral immunity, which is mediated by neutralizing VP1 antibody. Nearly all VP1 mutations in JCPyV-PML reside in one of the four solvent-exposed loops; these loops constitute the domains for binding the sialyated cell receptors and are the targets of the host's antibody response (*Buch et al., 2015*; *Lindner et al., 2019*; *Neu et al., 2010*). Thus, we asked if A2.V296F was resistant to neutralization by the mAb 8A7H5 (*Swimm et al., 2010*). Incubation of MuPyV with 8A7H5 prior to infection potently neutralized A2 but did not affect infectivity by A2.V296F. Replacement of V296 with alanine or isoleucine did not abrogate neutralization by 8A7H5, but substitution to tyrosine or tryptophan also conferred resistance to neutralization (*Figure 3A*). Spread of A2 infection in mouse fibroblast monolayers was significantly impaired by 8A7H5, but spread by A2.V296F was unimpeded (*Figure 3B*). To determine whether A2.V296F evaded neutralization by 8A7H5 in vivo, we passively immunized mice with 8A7H5 prior to s.c. infection and examined the efficacy of neutralization at day 4 or day 8 pi by measuring LT mRNA levels and the magnitude of the MuPyV-specific CD8 T cell response, respectively (*Figure 3C*). 8A7H5 immunization resulted in undetectable splenic LT mRNA levels in A2-infected mice 4 dpi, but had no effect on virus levels in A2.V296F-infected mice (*Figure 3D*); the identical pattern was seen for the anti-MuPyV CD8 T cell response (*Figure 3E*). Notably, we found that the IgG response to A2 MuPyV infection in WT mice competed with 8A7H5 for attachment to VP1, with 8A7H5 able to prevent binding of over 80% of MuPyV-specific IgG to

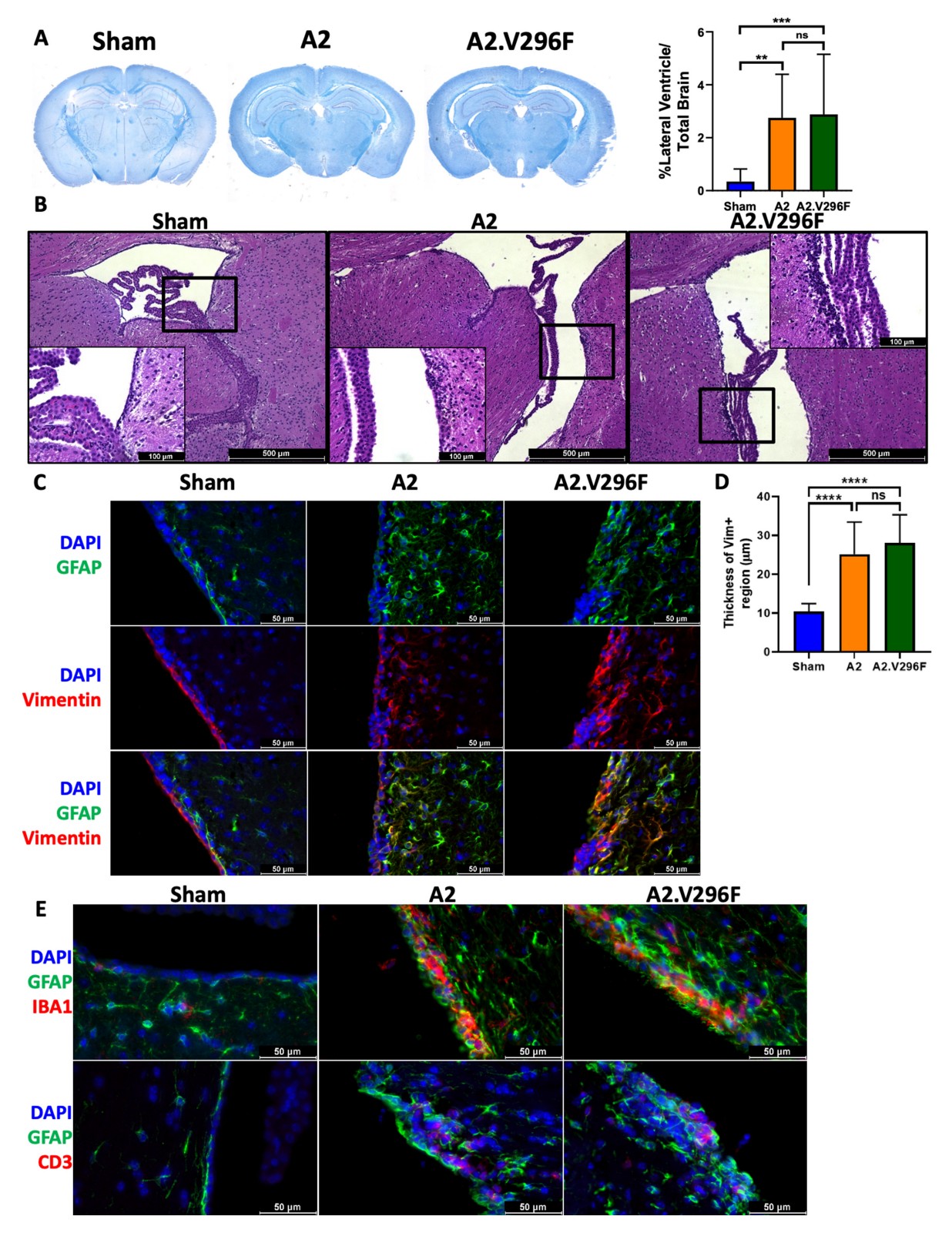

**Figure 2.** Persistent infection with either A2 or A2.V296F results in CNS pathology. (**A**) Left: LFB-PAS-stained brain sections 30 dpi with A2 or A2.V296F. Right: Hydrocephalus was quantified as the size of the lateral ventricle compared to total brain size. Data are from three independent experiments, n = 10–14 mice. For Sham vs A2 p=0.0022, Sham vs A2.V296F p=0.0010, and A2 vs. A2.V296F p=0.9806. (**B**) Representative 40x H & E images of the ependyma and choroid plexus of the lateral ventricle in mice 30 dpi. Inset image is 200x. (**C**) 400x fluorescence images of GFAP and vimentin

*Figure 2 continued on next page*

Figure 2 continued

expression in the lining of the lateral ventricles in mice 30 dpi. (D) Quantification of the thickness of the vimentin[+] region shown in (C). Data are from three independent experiments, n = 13–15 mice. For Sham vs. A2 p<0.0001, Sham vs. A2.V296F p<0.0001, and A2 vs. A2.V296F p=0.4626. (E) 400x fluorescence images of Iba1[+] and CD3[+] cells in the lateral ventricles 30 dpi. Data were analyzed by one-way ANOVA (A, D). **p<0.01, ***p<0.001, ****p<0.0001.

VP1 pentamers (*Figure 3F*). These findings suggested that a specific mutation in the sialic acid binding domains of VP1 diminished antibody neutralization of the virus.

## Cryo-EM reconstruction of MuPyV identifies mechanism of VP1 antibody escape by V296F

Similar to whole IgG, 8A7H5 Fab neutralizes and prevents spread of A2, but fails to do so for A2. V296F (*Figure 4—figure supplement 1*). To investigate how V296F conferred resistance to 8A7H5, we collected cryo-EM data for purified A2 capsids and A2 capsids incubated with saturating amounts of Fab (*Supplementary file 1*). Comparison of A2 and A2-Fab cryo-EM micrographs revealed an obvious difference in particle size, demonstrating successful formation of virus:Fab complexes (*Figure 4A*).

Refinement of both datasets produced maps at 3.9 Å and 4.2 Å resolution for the A2 and A2-Fab complex, respectively (*Figure 4B*, *Figure 4—figure supplement 2*, and *Supplementary file 2*; *Punjani et al., 2017*; *Zivanov et al., 2018*). In contrast to the 60 epitopes present in the BKPyV:scFv structure (one per asymmetric unit), our complex map revealed 360 epitopes were possible for 8A7H5 Fab, corresponding to six copies per asymmetric unit; that is, 1 Fab may bind each VP1 (*Figure 4—figure supplement 3*; *Lindner et al., 2019*). As a result, saturation with 8A7H5 Fab would effectively blanket the entire surface of the virus (*Figure 4B*). The central section through the complex map revealed Fab density approximately equal to that of the capsid, indicating most of the 360 available binding sites were occupied (*Figure 4C*).

After icosahedral refinement, we next proceeded with local sub-particle refinement of the constituent hexavalent and pentavalent capsomers for each dataset. This sub-particle refinement allows each capsomer additional degrees of freedom to move independently of the rigid icosahedral matrix, which compensates for imperfect icosahedral symmetry present in flexible virus capsids (*Goetschius et al., 2019*; *Stass et al., 2018*). Using ISECC, our custom implementation of the localized reconstruction approach, we computationally generated sub-particles from the refined whole particle images (*Figure 4—figure supplement 4*; *Abrishami et al., 2020*; *Ilca et al., 2015*). Sub-particle refinement improved the resolution of the pentavalent and hexavalent capsomers for both the A2 and A2-Fab complex (2.9 and 3.2 Å, respectively) (*Figure 4D and E*). For each dataset, the constituent capsomers attained approximately equal resolution (*Supplementary file 2*). This refinement also improved the resolution of the virus–Fab interface to 3.1–3.3 Å, with sidechain density clearly apparent (*Figure 4F* and *Figure 4—figure supplement 5*).

Given the improved resolution attained by sub-particle refinement, all atomic models were built in the sub-particle maps (*Supplementary file 2*). Virus models were initialized with a VP1 (PDB ID 1SIE) and Fab structures (PDB ID 3GK8) mutated to match the primary structure of the A2 strain virus and 8A7H5 Fab. The Fab structure required manually rebuilding the complementarity determining regions (CDRs). All models were then refined into cryo-EM density. Due to the quasi-equivalent VP1 molecules within the asymmetric unit (*Figure 4—figure supplement 3*), the six epitopes may have subtle conformational differences and during the build were not assumed to be identical. Therefore, one Fab was built into the pentavalent site and was subsequently docked into the remaining five sites on the hexavalent capsomer. After refining the build independently into each corresponding map density, loops comprising the six distinct epitopes in the asymmetric unit superimposed with a range of C alpha root mean square deviation (RMSD) of 0.40 Å to 0.51 Å. The resolution and map quality allowed us to identify unambiguously the 8A7H5 epitope and key contact residues (*Table 1*) for each of the six quasi-equivalent positions. Notably, these epitopes were found to be identical.

The main contact residues of the Fab mapped to the heavy chain with minor contributions from the light chain (*Supplementary file 3*). The heavy chain made all contacts with one copy of the coat protein, whereas the light chain interacted with the adjacent VP1 (*Figure 5—figure supplement 1*). On the virus capsid the 8A7H5 epitope consisted of thirteen residues of the BC and HI loops of VP1,

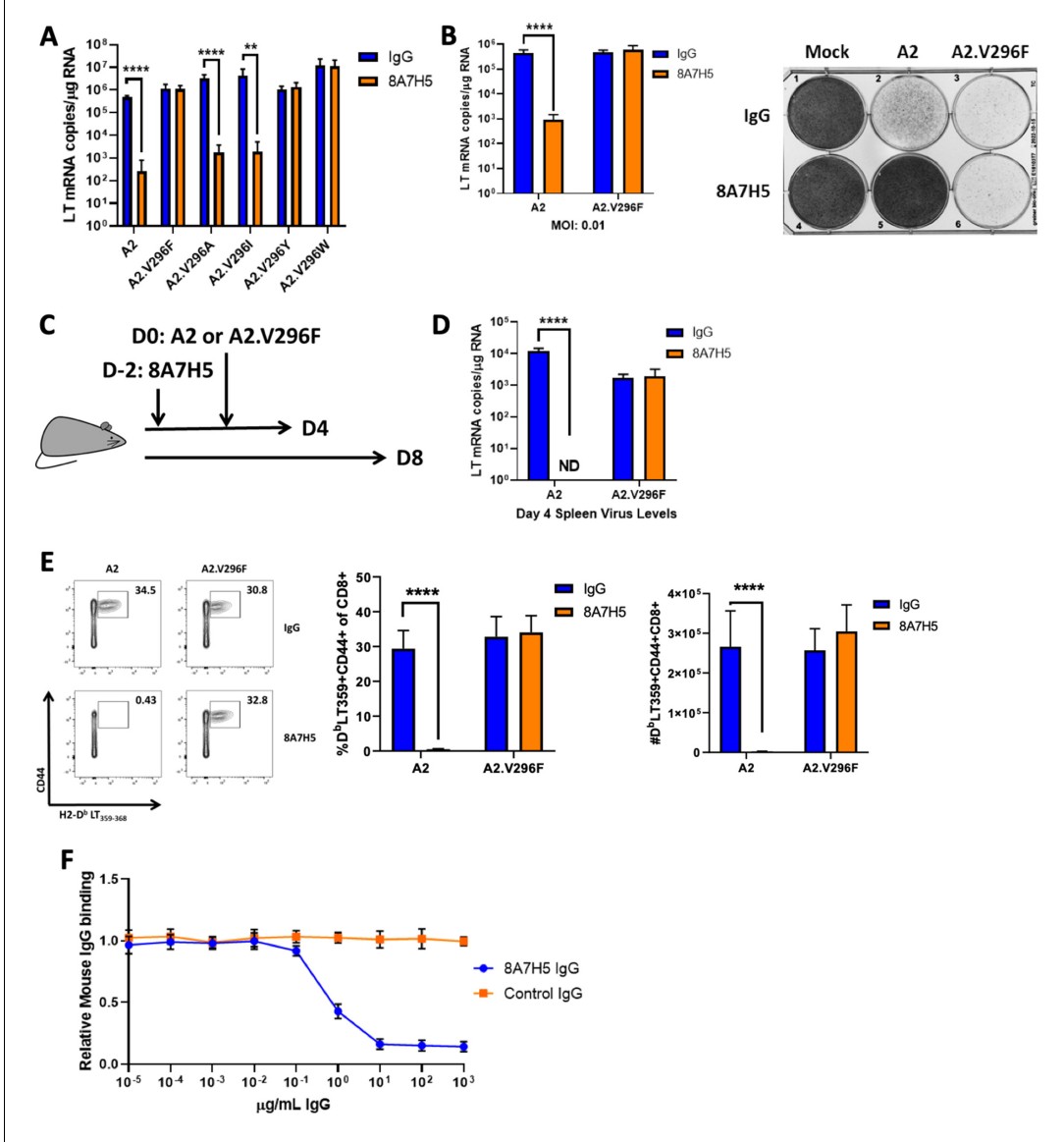

**Figure 3.** The V296F VP1 mutation confers resistance to a neutralizing mAb. (**A**) LT mRNA levels in NMuMG cells 24 hr pi with A2 or V296 mutant viruses preincubated with 8A7H5 or control IgG. Data are from two independent experiments, n = 12. For A2 p<0.0001, A2.V296F p=0.9626, A2.V296A p<0.0001, A2.V296I p=0.0076, A2.296Y p=0.5580, and A2.V296W p=0.9626. (**B**) Left: LT mRNA levels in A31 fibroblasts 96 hpi with A2 or A2.V296F at 0.01 MOI, 8A7H5 or control IgG was added to the media 24 hpi. Data are from two independent experiments, n = 6. For A2 p<0.0001 and A2.V296F p=0.3947. Right: Protection from virus-induced cell death. A31 fibroblasts were treated as in left panel, fixed with formaldehyde and stained with crystal violet 7 dpi. (**C**) Experimental design for in vivo neutralization experiments. (**D**) LT mRNA levels in the spleens of mice injected with 8A7H5 or control IgG followed by infection with A2 or A2.V296F. Data are from two independent experiments, n = 6 mice. For A2 p<0.0001 and A2.V296F p=0.7812. (**E**) Splenic T cell responses 8 dpi in mice treated and infected as in (**D**). Data are from two independent experiments, n = 6 mice. Middle: A2 p<0.0001, A2.V296F p=0.6758; Right: A2 p<0.0001, A2.V296F p=0.1679. (**F**) Competition for binding to VP1 pentamers between immune sera and 8A7H5 IgG. Sera from mice 30 dpi with A2 were diluted to 2 μg/mL of VP1-specific IgG and combined with increasing concentrations of 8A7H5 or control IgG. The serum/8A7H5 was then incubated with VP1 pentamers, and a mouse IgG-specific secondary antibody was used to measure the amount of serum bound to VP1 pentamers by ELISA. Each sample was normalized to binding in the absence of exogenous IgG. Data are from three independent experiments, n = 12 mice. Data were analyzed by multiple t tests (**A, B, D, E**). **p<0.01, ****p<0.0001.

and three residues from the DE (141, 151) and HI (292) loops of the adjacent VP1 (*Table 1*, *Figure 5A and B*). Notably the 8A7H5 epitopes map directly adjacent to one another in a ring tracing around the contours of the capsomer. There was a predicted salt-bridge between VP1 R77 and D99 located in CDR loop H3 of 8A7H5 (*Figure 5C*).

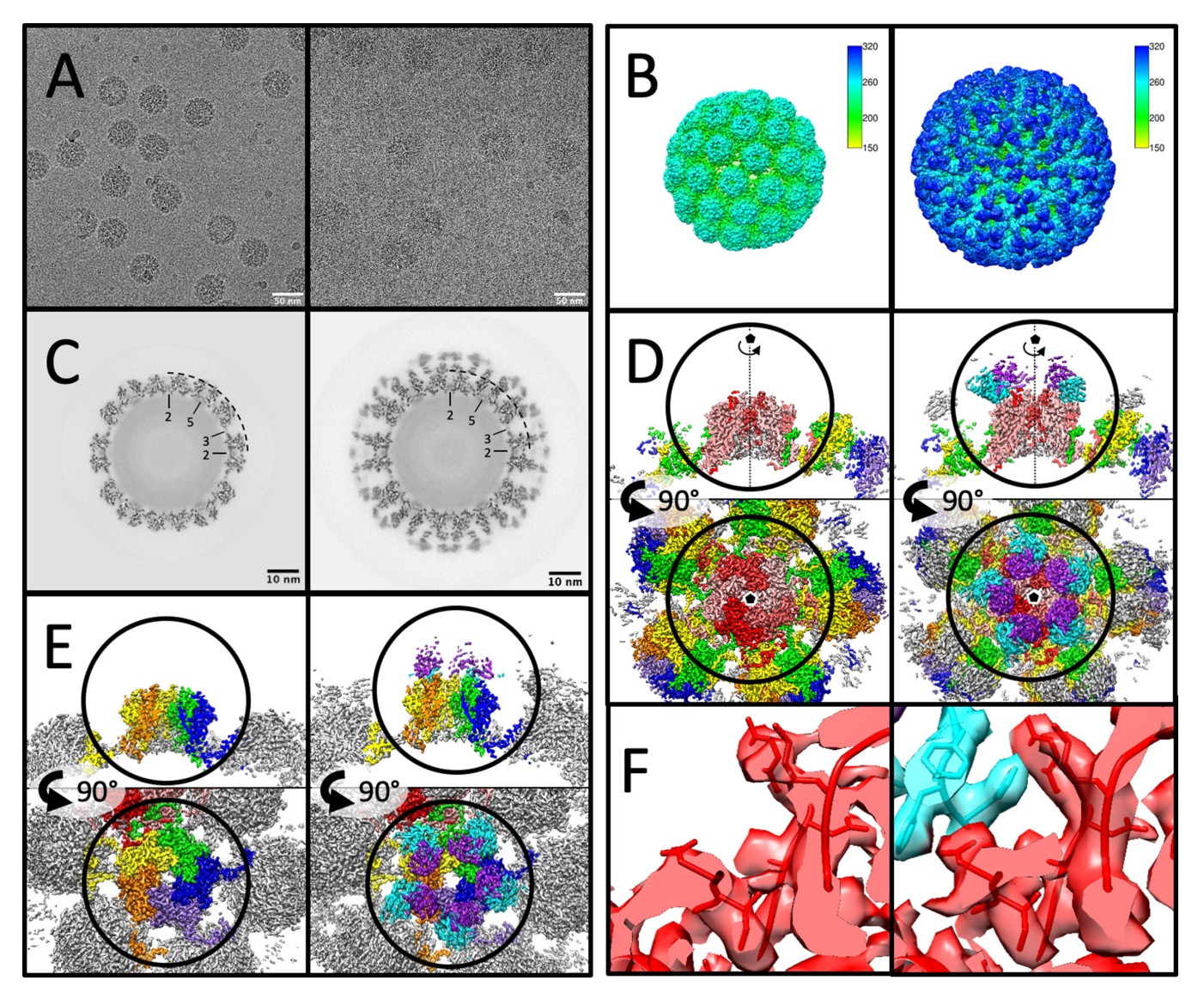

**Figure 4.** Cryo-EM image sub-particle refinement reconstructions showing architecture of MuPyV-Fab complexes. (**A**) Micrographs of virus and virus–Fab complex (shown left and right, throughout figure) illustrate particle diameter difference due to bound Fab. (**B**) Surface rendered icosahedrally averaged maps. (**C**) Central sections demonstrate the quality of the maps and show the Fab and capsid densities are of comparable magnitude. (**D** and **E**) Sub-particle refinement of pentavalent and hexavalent capsomers with sections through the maps (upper) and top-down views (lower) show the overall architecture. Pentavalent capsomers (D, VP1 density in shades of red) have fivefold symmetry (pentagon), whereas hexavalent capsomers have pseudo-symmetry (E, VP1 in OYGBV) most apparent in the contribution of VP1 C-terminal extensions to neighboring capsomers. Epitopes for the Fab molecules (light chain: purple; heavy chain: cyan) bridge adjacent VP1 molecules. (**F**) Local refinement of capsomer sub-particles resulted in interpretable sidechain density at the MuPyV-Fab interface (colors as in D and E).

The online version of this article includes the following figure supplement(s) for figure 4:

**Figure supplement 1.** 8A7H5 Fab neutralizes A2 but not A2.V296F.

**Figure supplement 2.** Fourier Shell Correlation (FSC).

**Figure supplement 3.** Asymmetric Unit.

**Figure supplement 4.** Refinement workflow.

**Figure supplement 5.** Local resolution.

**Table 1.** VP1 contact residues within −0.4 Å van der Waal's overlap.
The conformational epitope spans three loops over two copies of VP1. Contributions from the adjacent VP1 are denoted with '.

| Loop | Residue | |
| --- | --- | --- |
| BC | THR | 67 |
| | GLU | 68 |
| | ARG | 77 |
| | GLY | 78 |
| | ASN | 80 |
| | THR | 83 |
| | GLU | 91 |
| DE | PHE | 141' |
| | LYS | 151' |
| HI | ARG | 292' |
| | ASN | 293 |
| | TYR | 294 |
| | VAL | 296 |

The 8A7H5 epitope overlapped with residues associated with receptor binding (*Figure 5D*). This overlap suggests that the mechanism of antibody neutralization is to block the receptor-binding site and prevent virus interaction with the host cell. To address this experimentally, we analyzed the binding of H2B-GFP labeled A2 or A2.V296F to cells after pre-incubation of virus with 8A7H5. 8A7H5 mAb/Fab blocked A2 viral attachment, but had no effect on A2.V296F attachment as assayed by flow cytometry (*Figure 5E* and *Figure 5—figure supplement 2*). The V296F mutation placed a bulky phenylalanine sidechain directly within the Fab-virus interface likely disrupting the interaction with the CDR loop H3 through steric hindrance (*Figure 5F*). Consistent with this model, direct binding assays showed a significant reduction in 8A7H5 binding to A2.V296F compared to A2 (*Figure 5G* and *Figure 5—figure supplement 2*). The adjacent placement of epitopes resulted in a striking and tightly packed arrangement of bound Fab both within a capsomer (variable domain) and between capsomers (constant domain). Surprisingly, there was no steric clash observed between Fabs bound to neighboring epitopes (*Figure 5H* and *Figure 5—figure supplement 1*). This observation indicates that the MuPyV capsid was able to accommodate 360 copies of 8A7H5 Fab, saturating all available epitopes.

## Additional PML mutations in MuPyV impair kidney infection and disrupt 8A7H5 binding

Comparing the VP1 structures of JCPyV and MuPyV, we identified several additional substitutions to introduce into MuPyV to mimic other PML mutations (*Gorelik et al., 2011*; *Sunyaev et al., 2009*; *Figure 6A*). Of these mutations, only those at N293 and V296 resulted in viable MuPyV variants. All mutant viruses showed a defect in kidney infection following s.c. inoculation, similar to A2.V296F (*Figure 6B*). Inoculation i.c. with these viruses resulted in similar levels of infection in the brain compared to A2 (*Figure 6C*). The mutant viruses, however, varied in their susceptibility to 8A7H5-mediated neutralization. Predictably, V296Y conferred complete resistance to 8A7H5 (*Figure 3A*). A2.N293F and A2.N293Y remained sensitive to 8A7H5 neutralization (*Figure 6D*).

To examine the effect of the VP1 mutations that failed to produce virus, we generated recombinant VP1 pentamers with these mutations to test 8A7H5 binding. We further measured 8A7H5 avidity for VP1 by ELISA combined with NH4SCN treatment, a chaotropic agent that disrupts low-affinity antibody interactions (*Pullen et al., 1986*). 8A7H5 showed high avidity interactions with WT and N293Y VP1 pentamers, and low avidity with V296F pentamers. The R77N and H139R mutations each reduced 8A7H5 avidity to a similar extent as V296F, suggesting that these mutations would also confer resistance to 8A7H5. The T291D mutation also decreased 8A7H5 avidity, but not to the extent of

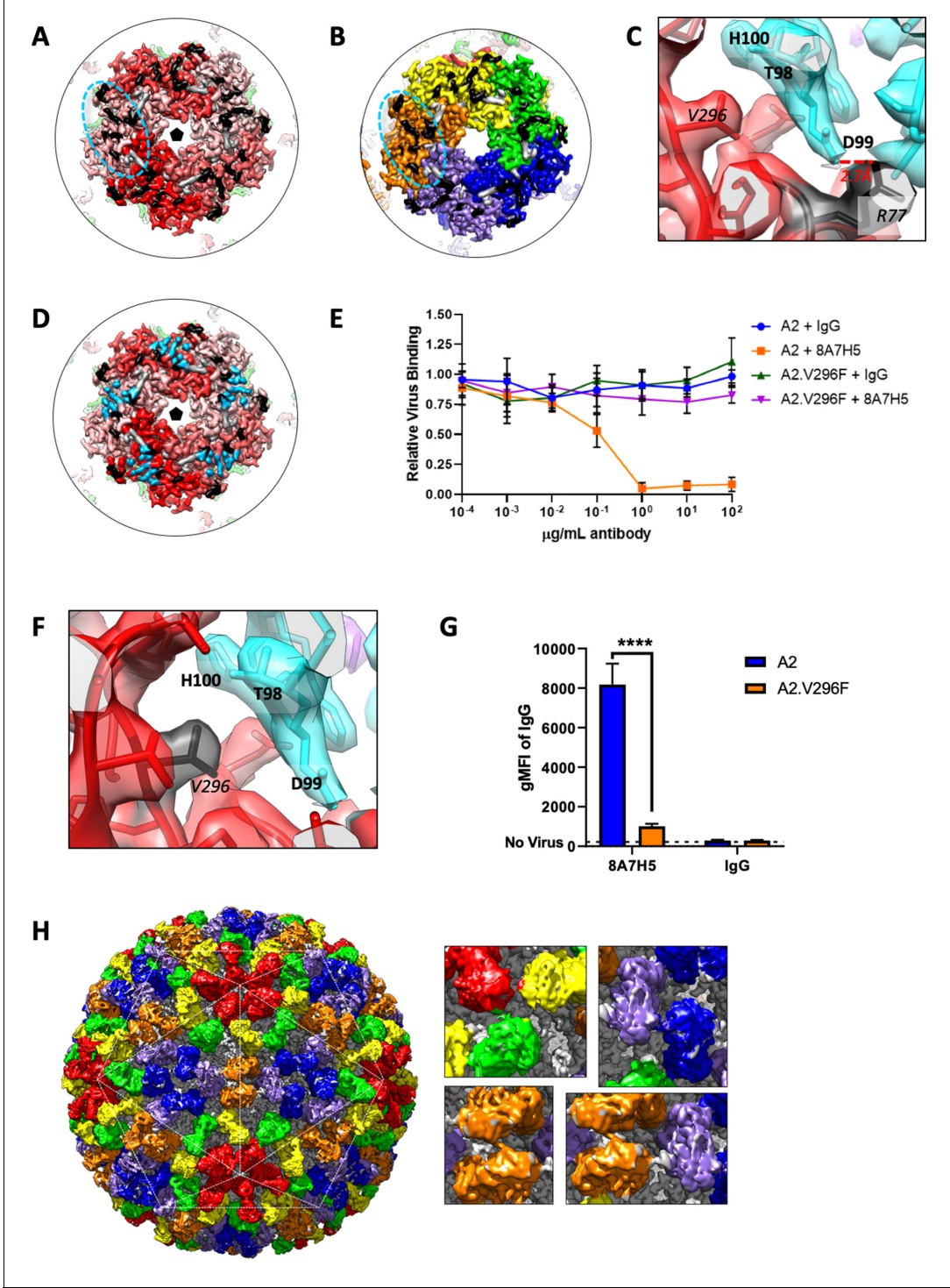

**Figure 5.** Cryo-EM reconstruction of MuPyV identifies mechanism of VP1 antibody escape by the V296F mutation. (**A and B**) The Fab epitope bridges adjacent copies of VP1 on the pentavalent capsomer (A, shades of red) and hexavalent capsomer (B, OYGBV). Neighboring epitopes abut directly against each other. Contact residues from the main VP1 chain are noted in black, with minor contributions from the adjacent VP1 in gray. (**C**) A salt-bridge is formed between R77 and Fab heavy chain residue D99. This interaction is near key residue V296, despite the large distance in linear sequence. (**D**) The Fab epitope and receptor-binding residues (sky blue) overlap (PDB ID 5CPY) (*Buch et al., 2015*). (**E**) Increasing concentrations of 8A7H5 prevent the attachment of A2, but not A2.V296F. H2B-GFP labeled virus was incubated with antibody prior to incubation with NMuMG cells. GFP fluorescence was measured by flow cytometry. Data are from two independent experiments, n = 6. (**F**) The V296F mutation would place a bulky residue at the MuPyV-Fab interface, disrupting the Fab heavy chain residue T98, D99, and H100 (cyan) interactions. (**G**) V296F prevents the binding of

*Figure 5 continued on next page*

Figure 5 continued

8A7H5 to VP1. NMuMG cells were incubated with A2 or A2.V296F followed by incubation with 8A7H5. Bound 8A7H5 was detected with an anti-IgG secondary. Data are from two independent experiments, n = 6 For 8A7H5 p<0.0001, IgG p=0.9693. (**H**) Six quasi-equivalent Fab molecules (red, orange, yellow, blue, green, purple) are contained within the asymmetric unit without clashes, despite the close proximity of Fab constant domains (inset). Data were analyzed by multiple t tests (**G**). ****p<0.0001.

The online version of this article includes the following figure supplement(s) for figure 5:

**Figure supplement 1.** Heavy and light chain interactions with the capsid surface.
**Figure supplement 2.** 8A7H5 Fab blocks A2 attachment and fails to bind A2.V296F.

R77N, H139R, and V296F (*Figure 6E* and *Figure 6—figure supplement 1*). Collectively, these data demonstrated that impaired kidney and retained brain tropism is a common theme for several PML mutations, but only a subset of these different mutations are capable of evading recognition by this VP1 mAb.

## 8A7H5 mAb selects VP1 escape mutations

We serially passaged A2 in the presence of 8A7H5 to select for de novo escape mutants. Three VP1 mutant viruses were isolated, an N80K point mutation in the VP1 BC loop and two single amino acid deletions, Δ294 and Δ295, adjacent to V296F in the HI loop. These mutations each conferred complete resistance to 8A7H5-mediated neutralization (*Figure 6—figure supplement 2*). VP1 residues N80, Y294, and D295 each map to or near contact residues predicted by the virus–Fab complex structure (*Figure 6F* and *Table 1*). Infection of mice s.c. with the mutant viruses resulted in elevated virus levels of A2.Δ294 and reduced virus levels of A2.N80K and A2.Δ295 in the kidney, indicating that these mutations have varying effects on kidney tropism (*Figure 6G*). All three mutant viruses had reduced brain infection levels following i.c. inoculation compared to A2 (*Figure 6H*). These differences in kidney tropism and decrease in brain tropism indicated that although these mutations shared 8A7H5-resistance with A2.V296F, individual mutations in this region had varying effects on tropism in an organ-specific manner.

## A2.V296F shows poor shedding under conditions of antibody escape

Our data suggested a model where PML-associated VP1 mutations promote antibody escape at the expense of infection and persistence in the kidney. This predicts that A2.V296F would be poorly shed in the urine, even under antibody-escape conditions in the host. μMT mice have a genetic defect in B cell development and fail to mount an anti-MuPyV antibody response (*Kitamura et al., 1991*; *Szomolanyi-Tsuda and Welsh, 1996*). To approximate the clinical observation of WT virus in the urine of PML patients despite mutant virus being in the blood and CSF, we inoculated μMT mice with a 1:1 ratio of A2:A2.V296F and began administering 8A7H5 four dpi. At 14 dpi, virus shed in the urine was heavily biased towards A2, despite the mice having high levels of A2.V296F in the blood and brain tissue (*Figure 6I*). This result showed that a severe impediment to kidney replication limits shedding of the V296F mutant virus in the urine, despite being viremic.

## Discussion

In this study, we elucidated the impact of MuPyV VP1 mutations on viral tropism and antibody neutralization, drawing a mechanistic link between JCPyV capsid mutations and PML pathogenesis. We applied a custom sub-particle refinement approach to reconstruct cryo-EM images of native capsid: antibody complexes at high resolution. The structures revealed the mechanism of VP1 antibody evasion. Using MuPyV with a VP1 mutation matching a frequent VP1 mutation in JCPyV-PML, we found that this viral variant retained tropism for the CNS, but was profoundly impaired in its ability to replicate in the kidney, a major organ reservoir for persistent polyomavirus infections. This mutation blocked neutralization by a MuPyV VP1 mAb via steric hindrance. Other JCPyV-PML VP1 mutations introduced into MuPyV also impaired kidney, but not brain infection, and varied in their ability to bind the VP1 mAb. mAb-escape MuPyV variants selected in vitro used additional mechanisms to evade neutralization but exhibited altered replication in both the brain and kidney. This disconnect between nAb escape and CNS tropism shows that only a subset of JCPyV VP1 variants refractory to

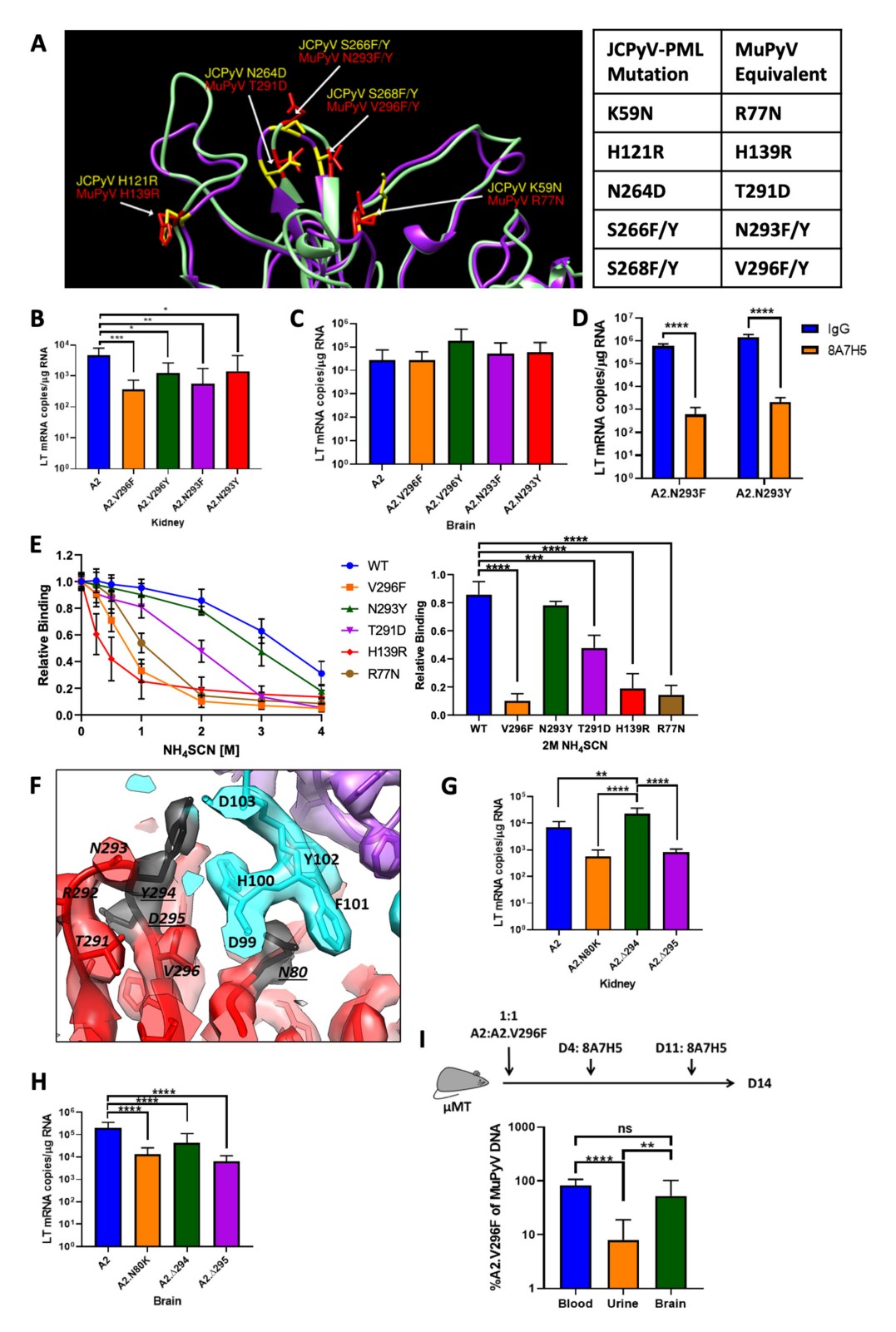

**Figure 6.** Additional JCPyV-PML mutations in MuPyV impair kidney, but not brain infection, and have varying effects on VP1 mAb neutralization. (**A**) Structural comparison of PML mutation sites in JCPyV VP1 (PDB 3NXG) with MuPyV VP1 (PDB 5CPU) residues (***Buch et al., 2015***; ***Neu et al., 2010***). (**B**) Kidney LT mRNA levels in mice 4 dpi with A2 or mutant viruses s.c. Data are from two independent experiments, n = 9–10 mice. For A2 vs. A2.V296F p=0.0008, A2 vs. A2.V296Y p=0.0100, A2 vs. A2.N293F p=0.0014, and A2 vs .A2.N293Y p=0.0168. (**C**) Brain LT mRNA levels in mice 4 dpi with A2 or

*Figure 6 continued on next page*

*Figure 6 continued*

mutant viruses i.c. Data are from four independent experiments, n = 13–14 mice. For A2 vs. A2.V296F p>0.9999, A2 vs. A2.V296Y p=0.2576, A2 vs. A2. N293F p=0.9974, and A2 vs. A2.N293Y p=0.9928. (**D**) LT mRNA levels in NMuMG cells 24 hpi with A2.N293F or A2.N293Y preincubated with 8A7H5 or control IgG. Data are from two independent experiments, n = 12. For A2.N293F p<0.0001 and A2.N293Y p<0.0001. (**E**) Analysis of 8A7H5 avidity for mutant VP1's using $NH_4SCN$. Left: Relative binding of 8A7H5 to recombinant VP1 with increasing concentrations of $NH_4SCN$. Right: Relative binding of 8A7H5 to recombinant VP1 at 2M $NH_4SCN$. Each point is the average of two technical replicates in an independent repeat, n = 3. For WT vs. V296F p<0.0001, WT vs. N293Y p=0.6440, WT vs. T291D p=0.0003, WT vs. H139R p<0.0001, and WT vs. R77N p<0.0001. (**F**) Location of novel changes in the VP1 escape mutations indicated by underlines. (**G**) LT mRNA levels 4 dpi in the kidneys of mice infected s.c. Data are from two independent experiments, n = 8 mice. For A2 vs. A2.N80K p=0.3374, A2 vs. A2.Δ294 p=0.0010, A2 vs. A2.Δ295 p=0.3724, A2.N80K vs. A2.Δ294 p<0.0001, A2.N80K vs. A2.Δ295 p=0.9999, and A2.Δ294 vs. A2.Δ295 p<0.0001. (**H**) LT mRNA levels 4 dpi in the brains of mice infected i.c. Data are from 2 to 4 independent experiments, n = 8–16 mice. For A2 vs. A2.N80K p<0.0001, A2 vs. A2.Δ294 p<0.0001, A2 vs. A2.Δ295 p<0.0001, A2.N80K vs. A2.Δ294 p=0.8811, A2. N80K vs. A2.Δ295 p=0.9979, and A2.Δ294 vs. A2.Δ295 p=0.6869. (**I**) Viral shedding in the urine is impaired by V296F, despite antibody escape in the blood. Top: Experimental design for infection of µMT mice and 8A7H5 treatment. Bottom: Frequency of A2.V296F DNA in blood, urine, and brain tissue of mice. Data are from three independent experiments, n = 13–14 mice. For Blood vs. Urine p<0.0001, Blood vs. Brain p=0.0500, and Urine vs. Brain p=0.0028. Data were analyzed by one-way ANOVA (**B, C, E** Right, **G, H, I**) or multiple t tests (**D**). In E, comparisons were only between WT and each mutant. **p<0.01, ***p<0.001, ****p<0.0001.

The online version of this article includes the following figure supplement(s) for figure 6:

**Figure supplement 1.** Relation of other VP1 mutations to virus–Fab interface.
**Figure supplement 2.** Resistance of escape mutations to 8A7H5 neutralization.

the VP1 antibody response are detected in PML patients. Our data support the concept that evasion of the VP1 antibody response facilitates the outgrowth of JCPyV variants capable of causing CNS injury.

Our implementation of sub-particle refinement allowed the rapid solution of the highest resolution cryo-EM structures of any polyomavirus map to date. This innovation was achieved using fewer particles compared to the traditional cryo-EM approach (*Supplementary file 4*). Improvements seen after sub-particle refinement may be attributable to capsid flexibility and the defocus gradient extending over the 45 nm diameter capsid. Curiously, correction for optics aberrations and the Ewald sphere effect improved resolution of only the icosahedrally averaged MuPyV capsid map, but not the MuPyV-Fab complex map (*Zivanov et al., 2020*). Resolution improvement from sub-particle refinement of MuPyV was equivalent to that seen with optics refinement, but these improvements were not cumulative. In contrast, the MuPyV-Fab complex map improved after sub-particle refinement. The lack of improvement from higher order aberration refinement for the complex map may be because refinement of optics parameters is a reference-based process, such that the flexibility contributed by 360 copies of Fab may be a barrier to solving and correcting properly for optical aberrations.

The densely packed arrangement of 360 Fab molecules coating the capsid (*Figure 5H*) signifies the presence of six structurally identical 8A7H5 epitopes within the asymmetric unit, despite the quasi-equivalence of VP1 molecules that form pentavalent and hexavalent capsomers. The conformational epitope bridges adjacent VP1 molecules within each capsomer, yet without provoking clash between neighboring Fab molecules. This binding pattern sharply contrasts with scFv 41F17, which recognizes a single structurally unique epitope within the BKPyV asymmetric unit (*Lindner et al., 2019*). The difference in binding behavior is due to the apical location of the 8A7H5 epitope that is comprised of structural features common to all capsomers. In contrast, the 41F17 epitope is laterally located and formed through the interaction of VP1 chains between adjacent hexavalent capsomers.

The structural data also explain antibody escape caused by VP1 mutations in JCPyV-PML when mapped into MuPyV. Introduction of bulky F/Y/W sidechains at position 296 within the antibody footprint likely promotes escape through steric collision, since an A or I residue at 296 retains sensitivity to neutralization. Loss of antibody binding to R77N pentamers is probably due to the lost saltbridge between VP1 R77 and D99 in CDR loop H3 of 8A7H5. T291D had an intermediate effect on 8A7H5 binding, consistent with its location immediately adjacent to several crucial contacts in the HI loop. Because H139 is not directly in the footprint, the mechanism of lost binding is not readily apparent but may be due to a long distance interaction. Although N293 (corresponding to JCPyV S266) is identified as a contact residue, it is on the periphery of the 8A7H5 footprint, which may explain retained recognition and neutralization of N293F/Y by 8A7H5. A recent report showed that

sera from healthy individuals and JCPyV sero-positive patients failed to recognize S266F but not wild type VP1 (*Jelcic et al., 2015*). Additionally, broadly neutralizing scFv 41F17 recognized an epitope comprised of residues from VP1 proteins between hexavalent capsomers (*Lindner et al., 2019*). Thus, mutations may disrupt recognition by antibodies with epitopes distinct from that of 8A7H5.

Our cryo-EM complex structures also explain three spontaneous escape mutations found during serial passage in the presence of 8A7H5. Deletion of contact residue Y294 or the immediately adjacent D295 provide escape through shortening and reorganization of the key antigenic HI loop. N80 directly interacts with 8A7H5 via a trio of residues (H100, F101, Y102) in CDR loop H3 of 8A7H5 that form a depression in the Fab topology into which the N80 side chain inserts (*Figure 6F*). The N80K substitution would disrupt this interaction by introducing a positive charge and a longer sidechain.

Several factors may lead to the strong neutralizing activity of 8A7H5 mAb. Because the 8A7H5 epitope bridges neighboring VP1 molecules within each capsomer, 8A7H5 binding may stabilize the virus and prevent uncoating. The salt-bridge formed between the R77 of VP1 on the virus and antibody residue D99 mimics the essential interaction of R77 with the sialic acid moiety of the host cell receptor (*Bauer et al., 1999*). The conformational epitope and the receptor-binding site both contain the HI loop; this significant overlap allows antibody binding to prevent attachment to the cellular receptor. 8A7H5 Fab recognizes 360 structurally identical epitopes on the virus capsid, despite the quasi-equivalence of the six VP1 chains within the asymmetric unit. There is no steric clash between the 360 copies of 8A7H5 Fab, resulting in the striking and tightly packed arrangement of Fab seen in *Figure 5H*. It is important to note that this packing would be unlikely to occur in vivo due to the bulk of a whole antibody.

JCPyV-PML VP1 mutations have been proposed to drive neurovirulence or evasion of humoral immunity, but not both (*Geoghegan et al., 2017*; *Jelcic et al., 2015*; *Maginnis et al., 2013*; *O'Hara et al., 2018*; *Ray et al., 2015*). Our data reconcile these findings, indicating that these mutations impair infection in sites of typical polyomavirus persistence (kidney, bone marrow), but retain infectivity in the CNS. We demonstrated that a MuPyV with the V296F PML-like VP1 mutation had profoundly impaired kidney tropism and lower viruria than parental MuPyV. Likewise, only archetype JCPyV is detected in urine, whereas VP1 mutant viruses are found in blood and CSF (*Gorelik et al., 2011*; *Reid et al., 2011*). This impaired kidney tropism by PML-VP1 mutants may underlie the absence of JCPyV-associated nephritis in PML patients, despite the kidney being the major site of JCPyV persistence (*Berger et al., 2017*).

Loss of kidney tropism and retained brain tropism by VP1 mutant viruses may be explained by differences in host cell receptor expression between these organs. Mutations in solvent-exposed VP1 loops could skew binding to a decoy receptor selectively expressed in the kidney that diverts virions to a nonproductive infection pathway. For example, the MuPyV VP1 mutation E91G impairs kidney infection by enabling attachment to branched chain in addition to straight chain sialyloligosaccharides (*Bauer et al., 1999*). Infection by the E91G mutant is also impaired by host cells expressing certain glycoproteins, which compete with glycolipid receptors guiding virion uptake into the productive infection pathway (*Qian and Tsai, 2010*). Alternatively, VP1 mutations could attenuate or even negate binding to receptor(s) necessary for kidney infection, whereas a different receptor(s) is (are) expressed in the CNS. Supporting this possibility is recent evidence showing that WT JCPyV can bind both sialyated glycans and non-sialyated glycosaminoglycans (GAGs), whereas JCPyV-PML VP1 mutants only bind GAGs (*Geoghegan et al., 2017*). Productive kidney infection may depend on virus binding to sialyated glycans but brain infection may also use GAG receptors, which would then enable both WT and VP1 mutant JCPyVs to infect glial cells (*Kondo et al., 2014*).

Emergence of VP1 mutant JCPyVs in PML patients but not healthy individuals infers that viruses with these mutations have a replication advantage in the setting of depressed immune status (*Zheng et al., 2005*). Mutations in the four receptor-binding loops of VP1 are typically detrimental to viral fitness and persistence (*Bauer et al., 1999*; *Caruso et al., 2003*), and our data showing altered tropism by MuPyV VP1 mutants are clearly aligned with this idea. Evasion of host antibodies provides a strong selective pressure to promote the spread of an otherwise replication-disadvantageous mutation. Experimental demonstration of this scenario comes from evidence that the mutant A2.V296F virus strongly outcompeted parental A2 virus in the blood and brain, but was still poorly shed into the urine, when faced with an A2-nAb (*Figure 6I*). Our data agree with recent reports showing poor neutralization by sera from PML patients for their VP1 mutant JCPyVs, and indicate

that selection of VP1 mutants is driven by an antibody response sufficient to control parental but not a VP1 mutant virus (*Jelcic et al., 2015*; *Ray et al., 2015*). Thus, our findings indicate that JCPyV takes a hit to viral fitness in order to evade humoral immunity.

By extension, our results strongly support the concept that antibody escape is a requisite first step in PML development. The resulting viremia, then, would precede viral entry into the brain, whether by infiltrating the CSF via the choroid plexus, direct infection of brain endothelium, or by hitchhiking a cellular vehicle (*Chapagain et al., 2007*; *Dörries et al., 2003*; *Houff et al., 1988*; *von Einsiedel et al., 2004*). JCPyV viremia is found in multiple sclerosis patients treated with natalizumab (*Major et al., 2013*). In support of a choroid plexus-mediated route, JCPyV infects primary choroid plexus epithelial cells, and JCPyV-infected choroid plexi are found in PML brains (*Corbridge et al., 2019*; *O'Hara et al., 2020*; *O'Hara et al., 2018*). Both A2 and A2.V296F viruses productively infect the ependyma, and we reported ependymal infection by MuPyV under conditions of immune suppression (*Mockus et al., 2020*). Infection of the choroid plexus and ependyma may serve as a viral staging area for JCPyV invasion of the brain parenchyma, providing a foothold for viral dissemination in the CNS parenchyma.

A2 and A2.V296F induced comparable bilateral dilatation of the lateral ventricles (*Figure 2A and B*). Ventricular enlargement has been reported late in the course of PML disease and in a case of JCPyV meningitis (*Agnihotri et al., 2014*; *Ray et al., 2015*). Loss of periventricular tissue due to infection/inflammation could lead to a compensatory ventricular enlargement rather than obstructive hydrocephalus. Alternatively, increased ventricular volume in the infected brain could result from transudation of plasma from a leaky choroid plexus, increased CSF production, or obstructed CSF flow.

Using the MuPyV CNS infection model, we demonstrate that evasion of the host's neutralizing antiviral humoral response is the dominant driver of VP1 mutant viruses that retain CNS tropism. We developed a custom sub-particle refinement approach to reconstruct efficiently cryo-EM structures of polyomavirus capsid-Fab complexes at the highest resolution to date. These structures elucidated the mechanisms of neutralization and antibody escape. Our findings argue against the concept that VP1 mutations act per se to render JCPyV neurovirulent. Instead our work supports the model that viremia, consequent to outgrowth of antibody-escape VP1 variants, is a critical step in PML pathogenesis.

## Materials and methods

**Key resources table**

| Reagent type (species) or resource | Designation | Source or reference | Identifiers | Additional information |
|---|---|---|---|---|
| Antibody | Anti-VP1 (Rat Clone 8A7H5) | *Swimm et al., 2010* | Clone 8A7H5 | See Materials and methods for concentrations |
| Antibody | ChromPure Rat IgG | Jackson Immuno Research | Cat#012-000-003 | See Materials and methods for concentrations |
| Antibody | Anti-CD8β (Rat monoclonal) | *Pierres et al., 1982* | Clone H35-17.2 | 250 µg per injection |
| Antibody | Anti-VP1 (Rabbit polyclonal) | Provided by Robert Garcea (University of Colorado Boulder) | | IF(1:1000) |
| Antibody | Anti-Vimentin (Rat monoclonal) | R & D Systems | Cat#MAB2105 | IF(1:100) |
| Antibody | Anti-GFAP (Goat polyclonal) | Abcam | Cat#ab53554 | IF(1:1000) |
| Antibody | Anti-Iba1 (Rabbit polyclonal) | FUJIFILM Wako | Cat#019–19741 | IF(1:500) |
| Antibody | Anti-CD3 (Rabbit monoclonal) | Abcam | Cat#ab16669 | IF(1:100) |
| Antibody | Anti-Goat IgG AF488 (Bovine polyclonal) | Jackson Immuno Research | Cat#805-545-180 | IF(1:500) |

*Continued on next page*

*Continued*

| Reagent type (species) or resource | Designation | Source or reference | Identifiers | Additional information |
|---|---|---|---|---|
| Antibody | Anti-Rat IgG AF568 (Donkey polyclonal) | Abcam | Cat#ab175475 | IF(1:500) |
| Antibody | Anti-Rabbit IgG AF647 (Donkey polyclonal) | Jackson Immuno Research | Cat#711-605-152 | IF(1:500) |
| Antibody | Anti-CD8α-AF700 (Rat monoclonal) | Biolegend | Cat#100730 | FC(1:200) |
| Antibody | Anti-CD44-FITC (Rat monoclonal) | Biolegend | Cat#103006 | FC(1:200) |
| Antibody | Anti-Rat IgG-APC (Goat polyclonal) | BD | Cat#551019 | FC(1:200) |
| Antibody | Anti-Mouse IgG-HRP (Goat polyclonal) | Biolegend | Cat#405306 | ELISA(1:2800) |
| Antibody | Anti-Mouse IgG-HRP (Goat polyclonal) | Bethyl Laboratories INC | Cat#A90-116P | ELISA(1:7000) |
| Antibody | Biotinylated Anti-Rabbit (Goat Polyclonal) | Vector Laboratories | Cat#BA-1000 | IHC(1:500) |
| Other | Mouse Polyomavirus (Strain A2) | N/A | N/A | |
| Strain, strain background (*Escherichia coli*) | BL21 | Agilent | Cat#200133 | |
| Recombinant DNA reagent | PyVP1-pGEX-4T-2 (plasmid) | Provided by Robert Garcea | N/A | |
| Recombinant DNA reagent | H2B-GFP (plasmid) | *Kanda et al., 1998*, Addgene | Plasmid #11680 | |
| Peptide, recombinant protein | Benzonase Nuclease | Sigma | Cat#E1014 | Virus Purification (1:3333) |
| Other | $D^b$-LT359 Tetramer | NIH Tetramer Core | N/A | FC(1:400) |
| Peptide, recombinant protein | Neuraminidase from *Vibrio cholerae* (Type II) | Sigma | Cat#N6514 | Virus Purification (1:2000) |
| Peptide, recombinant protein | RevertAid H Minus Reverse Transcriptase | ThermoFisher | Cat#EP0451 | |
| Chemical compound, drug | OptiPrep | STEMCELL Technologies | Cat#07820 | |
| Chemical compound, drug | Glutathione Sepharose 4B | GE Healthcare | Cat#17075601 | |
| Chemical compound, drug | TRIzol Reagant | ThermoFisher | Ref#15596018 | |
| Chemical compound, drug | Lipofectamine 2000 Transfection Reagent | ThermoFisher | Cat#11668030 | |
| Commercial assay, kit | TBP PrimeTime XL qPCR Assay | IDT | Mm.PT.39a.22214839 | |
| Other | ProLong Gold antifade reagent with DAPI | ThermoFisher | Ref#P36931 | |
| Other | Fixable Viability Dye eFluor780 | ThermoFisher | Cat# 65-0865-14 | FC(1:1000) |
| Commercial assay, kit | Avidin/Biotin Blocking Kit | Vector Laboratories | Cat#SP-2001 | |
| Commercial assay, kit | VECTASTAIN Elite ABC-HRP Kit | Vector Laboratories | Cat#PK-6100 | |
| Commercial assay, kit | NovaRED Substrate Kit | Vector Laboratories | Cat#SK-4800 | |

*Continued on next page*

*Continued*

| Reagent type (species) or resource | Designation | Source or reference | Identifiers | Additional information |
|---|---|---|---|---|
| Commercial assay, kit | 1-Step Ultra TMB-ELISA | ThermoFisher | Ref#34028 | |
| Commercial assay, kit | PerfectCTa SYBR Green FastMix | Quantabio | P/N 84069 | |
| Commercial assay, kit | PerfectCTa FastMix II ROX | Quantabio | P/N 84210 | |
| Commercial assay, kit | PureLink Viral RNA/DNA mini Kit | ThermoFisher | Ref#12280–050 | |
| Commercial assay, kit | Pierce Fab Micro Preparation Kit | ThermoFisher | Ref#44685 | |
| Commercial assay, kit | Nab Protein G Spin Columns | ThermoFisher | Ref#89953 | |
| Commercial assay, kit | Wizard Genomic DNA Purification Kit | Promega | Ref#A1120 | |
| Commercial assay, kit | QuikChange II Site-Directed Mutagenesis Kit | Agilent | Cat#200523 | |
| Commercial assay, kit | QIAquick PCR Purification Kit | Qiagen | Cat#28104 | |
| Cell Line (*Mus musculus*) | BALB/3T3 Clone A31 | ATCC | CCL-163, RRID:CVCL_0184 | |
| Cell Line (*M. musculus*) | NMuMG | ATCC | CRL-1636, RRID:CVCL_0075 | |
| Cell Line (*M. musculus*) | mIMCD-3 | ATCC | CRL-2123, RRID:CVCL_0429 | |
| Cell Line (*M. musculus*) | C57BL/6 MEF | This paper | | Primary murine embryonic fibroblasts |
| Strain, strain background (*M. musculus*) | C57BL/6 | National Cancer Institute | Cat#OIC55 | |
| Genetic reagent (*M. musculus*) | $Stat1^{-/-}$ | Jackson Laboratory | Cat#012606 | |
| Genetic reagent (*M. musculus*) | µMT | Jackson Laboratory | Cat#002288 | |
| Software, algorithm | Prism | Graphpad | RRID:SCR_002798 | |
| Software, algorithm | FlowJo | BD | RRID:SCR_008520 | |
| Software, algorithm | ImageJ | NIH | RRID:SCR_003070 | |
| Software, algorithm | Leica LAS X | Leica | RRID:SCR_013673 | |
| Software, algorithm | Photoshop | Adobe | RRID:SCR_014199 | |
| Software, algorithm | Relion | *Scheres et al., 2009* | RRID:SCR_016274 | |
| Software, algorithm | cryoSPARC | Structura Biotechnology | RRID:SCR_016501 | |
| Software, algorithm | ISECC | See Data and code availability | v 2019.09 | |
| Software, algorithm | PHENIX | phenix-online.org | RRID:SCR_014224 | |
| Software, algorithm | Coot | *Emsley et al., 2010* | RRID:SCR_014222 | |

## Mice

C57BL/6 mice were purchased from the National Cancer Institute and μMt mice were purchased from the Jackson Laboratories. $Stat1^{-/-}$ mice (The Jackson Laboratory) were kindly provided by Dr. Christopher Norbury (Penn State College of Medicine). Male and female mice were used for experiments between 6–15 weeks of age. Mice of the same sex/age were randomly assigned to experimental groups. Mice were housed and bred in accordance with the National Institutes of Health and AAALAC International Regulations. The Penn State College of Medicine Institutional Animal Care and Use Committee approved all experiments.

## Virus strains

All work was performed with the A2 strain of MuPyV. Viral stocks were generated by transfection of viral DNA into NMuMG cells using Lipofectamine 2000 Transfection Reagent (ThermoFisher). A single passage in NMuMG cells was used for viral amplification to generate a high titer virus stock. Virus stocks were titered on A31 fibroblasts by plaque assay (*Lukacher and Wilson, 1998*).

## Cell lines and primary cells

The 8A7H5 hybridoma (rat IgG2b, κ) was previously generated by immunization of rats with MuPyV VP1 virus-like particles (*Swimm et al., 2010*). NMuMG, BALB/3T3 clone A31 'A31', and mIMCD-3 cells were purchased from ATCC. Mouse embryonic fibroblasts (MEFs) were isolated from day 13 C57BL/6 embryos. Hybridomas 8A7H5 and H35-17.2 (anti-CD8β) (*Pierres et al., 1982*) were maintained in PFHM-II Protein-Free Hybridoma Medium (ThermoFisher) at 37°C in 5% $CO_2$. mAb was generated by growing the hybridomas in CELLine disposable bioreactor flasks (Corning). All other cells were maintained in Dulbecco's Minimal Eagle Media supplemented with 10% fetal bovine serum, 100 U/mL penicillin, and 100 U/mL streptomycin (DMEM) at 37°C in 5% $CO_2$. The sex of NMuMG cells is female, the sex of mIMCD-3 and A31 cells is not reported. Cell lines were mycoplasma negative, authenticated by STR profiling (ATCC), examined for correct cell morphology, and used at low passage number.

## Generation of mutant viruses

Viral mutants were generated by site-directed mutagenesis using the Quikchange II Site-directed mutagenesis kit (Agilent) with forward and reverse primers specific for each mutation (*Supplementary file 6*). To confirm the presence of the mutations, viral DNA was isolated from virus stocks and the VP1 region was PCR amplified and sequenced.

## Virus infections

Mice were infected with MuPyV s.c. via the hind footpad with $1 \times 10^6$ PFU or i.c. with $5 \times 10^5$ PFU. For CD8 T cell depletions, mice received 250 μg of anti-CD8β in PBS intraperitoneally (i.p.) three and one days prior to infection. For in vivo neutralization experiments, mice received 250 μg of 8A7H5 or control IgG in PBS i.p. on the specified days. For in vitro experiments, subconfluent cells were incubated with virus for 1.5 hr at 4°C, and then free virus was removed by washing with DMEM. For single cycle replication and plaque assays, free virus was not removed. Following infection, cells were maintained in DMEM at 37°C in 5% $CO_2$.

## Viral genome quantification

50 μL of viral lysate was treated with 250 U of Benzonase Nuclease (Sigma) in 250 μM $MgCl_2$ at 37°C for 1 hr. Viral genomes were then isolated using the Invitrogen Purelink Viral RNA/DNA Mini Kit (ThermoFisher Scientific). Viral genomes were quantified by Taqman qPCR with primers and probe targeted to the LT region of the viral genome (*Supplementary file 6*; *Wilson et al., 2012*).

## Infection neutralization assay

10 μg of 8A7H5 mAb/Fab or control IgG/Fab (Jackson ImmunoResearch) was incubated at 4°C for 30 min with $1 \times 10^4$ PFU of MuPyV and then added to $1 \times 10^5$ NMuMG cells. Cells were infected at 4°C for 1.5 hr and mRNA was harvested 24 hr later. To measure antibody inhibition of viral spread, A31 cells were infected at an MOI of 0.01 and 8A7H5 or IgG-containing media (10 μg/mL) was added 24 hpi. Viral spread was quantified by mRNA at 96 hpi or crystal violet staining nine dpi.

## Viral mRNA quantification

RNA was harvested with TRIzol Reagent (ThermoFisher) and isolated by phenol:chloroform extraction followed by isopropanol precipitation. cDNA was prepared with 1–2 ug of RNA using random hexamers and Revertaid RT (ThermoFisher). LT mRNA levels were quantified by Taqman qPCR with normalization to TATA-Box Binding Protein (IDT) and compared to a standard curve (*Maru et al., 2017*).

## H2B-GFP labeling of MuPyV and 8A7H5 mAb attachment assays

Virus was labeled by infection of NMuMG cells expressing an H2B-GFP fusion protein, which is incorporated into the PyV minichromosome during DNA replication and packaging (*Fang et al., 2010*; *Geoghegan et al., 2017*; *Kanda et al., 1998*). 8A7H5 mAb or control IgG was incubated with labeled A2 or A2.V296F at a ratio of 5000 encapsidated viral genomes/cell for 30 m at 4°C, then added to a suspension of $5 \times 10^4$ trypsinized NMuMG cells and incubated for 30 m at 4°C. Cells were then washed twice in PBS and fixed for 20 m in 2% PFA. GFP fluorescence on the cells was quantified using a BD LSRFortessa Flow Cytometer and normalized to fluorescence of virus bound in the absence of antibody. To measure 8A7H5 attachment to virions, virus was incubated with cells for 30 m at 4°C, then treated with 8A7H5. Bound 8A7H5 was stained with APC anti-rat IgG and quantified using a BD LSRFortessa Flow Cytometer.

## Fab generation and mAb sequencing

8A7H5 and control Fabs were generated using the Pierce Fab Micro Preparation Kit (Thermo Fisher) and purified on Protein G columns (Thermo Fisher). Sequencing of the heavy and light chains of the mAb was carried out as previously described (*Guan et al., 2015*). In brief, hybridoma cells were pelleted and RNA was extracted with TRIzol Reagant (ThermoFisher). cDNA was generated with Revertaid RT (ThermoFisher) and amplified by PCR using *PfuTurbo* DNA Polymerase (Agilent) with published primers (*Wang et al., 2000*). PCR products were purified using the QIAquick PCR Purification Kit (Qiagen) and sequenced.

## Virus purification for Cryo-EM

Virus purification was adapted from a published BKPyV purification method (*Hurdiss et al., 2018*). NMuMG cells were infected at low MOI with A2 or A2.V296F. Following cell lysis, media/lysate was collected and cell debris was pelleted at 15,000 g for 20 m. The supernatant was collected, and the pellet was resuspended in Buffer A (10 mM Hepes, 1 mM CaCl$_2$, 1 mM MgCl$_2$, 5 mM KCl). The pellet was freeze-thawed three times, treated with Benzonase Nuclease (75 U/mL) (Sigma) and type II neuraminidase (1/2000) (Sigma) for 1 hr at 37°C, then combined with 0.1% deoxycholic acid and incubated at 37°C for 15 m, followed by 42°C for 5 m. The sample was pelleted at 15,000 g for 20 m, and the supernatant was collected and combined with the original supernatant. The combined supernatants were layered on a 20% sucrose cushion and spun for 3 hr at 85,000 g. The pellet was resuspended in Buffer A and layered on top of a 27/33/39% gradient of OptiPrep (STEMCELL) in Buffer A. The sample was spun at 237,000 g for 3.5 hr at 16°C. The band containing the virus was then removed with a syringe.

## VP1 pentamers

Full-length MuPyV VP1 in the pGEX-4T-2 expression plasmid was provided by Robert Garcea (University of Colorado, Boulder). VP1 mutants were generated using the Quikchange II Site-directed mutagenesis kit (Agilent) with forward and reverse primers listed (*Supplementary file 6*). VP1 pentamers were induced by IPTG in BL21 *E. coli* (Agilent), and purified with glutathione sepharose (GE Healthcare) followed by thrombin cleavage. Pentamers were then bound and eluted from a cellulose phosphate column.

## ELISA

ELISAs were performed using 50 ng of VP1 pentamer/well in an EIA/RIA Polystyrene High Bind Microplate (Fisher Scientific) coated overnight at 4°C. For 8A7H5 competition with immune mouse sera, the VP1-specific IgG concentration of the serum was measured, then 100 ng of VP1-specific IgG was combined with increasing concentrations of 8A7H5 or control IgG for the ELISA. Bound

mouse IgG was detected with a mouse IgG-specific secondary (Biolegend). For avidity measurements, 8A7H5-pentamer complexes were treated with $NH_4SCN$ in 0.1 M phosphate for 15 m before detection of 8A7H5 mAb. For each VP1 variant, 8A7H5 binding was normalized to signal in the absence of $NH_4SCN$.

### Flow cytometry

Single cell suspensions of splenocytes were stained with antibody/tetramer cocktails in 100 μL for 30 m and quantified using a BD LSRFortessa Flow Cytometer. Flow cytometry data were analyzed using FlowJo software (BD).

### mAb-mediated selection for VP1 escape mutants

One x $10^5$ NMuMG cells were infected with A2 at an MOI of 0.1. Twenty-four hpi, 0.5 μg/mL 8A7H5 was added to the media. The medium was replaced every 3–4 days until cell death 1–2 weeks post-infection. The lysate was collected and diluted 1/100 to infect new NMuMG cells. After 3–4 passages, the resulting lysate was diluted 1/100 and combined with 10 μg of 8A7H5 for 30 m prior to infection of NMuMG cells. Following infection, the cells were maintained in 10 μg/mL 8A7H5 and observed for cell death/lysis. Lysates were then collected and viral DNA was isolated using the Pure-Link Viral RNA/DNA mini Kit (ThermoFisher). The VP1 region of the genome was then amplified by PCR and sequenced. Identified mutants were cloned or generated by site-directed mutagenesis and confirmed to be escape mutants by neutralization assay.

### Immunofluorescence microscopy, immunohistochemistry, and histology

Mice were perfused with 10% heparin in PBS, followed by 10% neutral buffered formalin (NBF). Heads were fixed overnight in NBF and brains were then removed, paraffin-embedded, and sectioned. Kidneys were removed from mice and fixed overnight in 10% NBF before paraffin-embedding and sectioning. For histology, sections deparaffinized and were stained with hematoxylin and eosin (H & E) or Luxol Fast Blue-Periodic Acid Schiff (LFB-PAS). For immunofluorescence and VP1 immunohistochemistry (IHC), sections were deparaffinized and subjected to antigen retrieval (95°C in 10 mM sodium citrate pH 6 for 10 m). Sections were permeabilized with 1% TritonX-100 for 15 m and then washed 2x in PBST (0.1% Triton X-100, 0.05% Tween20). For IHC, sections were incubated in 0.3% $H_2O_2$ in PBST for 30 m, then blocked for avidin and biotin for 15 m each (Vector). Sections were blocked with blocking buffer (5% BSA in PBST) for 2 hr and incubated overnight at 4°C with primary antibodies in blocking buffer. Sections were washed 3x with PBST, incubated with secondary antibodies in blocking buffer for 1.5 hr, washed 3x with PBST, then sections stained with fluorophore-conjugated antibodies were mounted with ProLong Gold Antifade with DAPI (Thermofisher). For VP1 IHC, sections were incubated with VECTASTAIN Elite ABC-HRP (Vector) for 10 m, washed 3x with PBST, then developed with the Vector NovaRED peroxidase substrate kit (Vector). Sections were then counterstained with hematoxylin and mounted. LFB images were acquired on a Keyence BZ-X710 all-in-one fluorescence microscope; immunofluorescence and histology images were acquired on a Leica DM4000 fluorescent microscope. The thickness of the vimentin[+] region was quantified across six images/sample in ImageJ (NIH). Hydrocephalus was quantified by measuring the pixel area of right and left lateral ventricles and dividing by the pixel area of the total brain section (*Mockus et al., 2020*). For representative fluorescence and brightfield images, adjustments for brightness/contrast were done uniformly to all images in the group using LAS X (Leica).

### DNA isolation and quantification

Solid tissues were homogenized using a TissueLyser II (Qiagen). DNA was isolated using the Wizard Genomic DNA Purification Kit (Promega). DNA was isolated from lysates, blood, and urine using the PureLink Viral RNA/DNA mini Kit (ThermoFisher). For competition experiments, total viral DNA and V296F DNA was quantified by Sybr Green qPCR (Quantabio) using the LT DNA and V296F DNA qPCR primers, respectively (*Supplementary file 6*). The ratio of A2:A2.V296F was determined by comparison to a standard curve of known A2:A2.V296F DNA ratios.

### Cryo-EM and data collection

MuPyV was buffer exchanged against 10 mM HEPES pH 7.9, 1 mM CaCl$_2$, 1 mM MgCl$_2$, 5 mM KCl (*Hurdiss et al., 2016*). MuPyV (2.8 mg/mL) was incubated with 8A7H5 Fab (1.1 mg/mL) for 30 m at room temperature. For vitrification of each sample, a 3 μL aliquot was applied to a freshly glow-discharged QUANTIFOIL EM grid. Grids were blotted for 3 s in 95% relative humidity before plunging into (Vitrobot; Thermo Fisher) liquid ethane. Cryo-EM datasets were collected at 300 kV with a Titan Krios microscope (Thermo Fisher) equipped with a spherical aberration corrector at the Huck Institute for Life Sciences cryo-EM Facility. Automated single-particle data acquisition was performed with EPU using the Falcon three detector with a nominal magnification of 59,000x, yielding a final pixel size of 1.1 Å/pixel (*Supplementary file 1*).

### Image processing

Patch motion correction, patch-CTF estimation, particle picking, 2D classification, and icosahedral refinement were performed in cryoSPARC (*Punjani et al., 2017*). Particles transferred to RELION version three for polishing, before another round of icosahedral refinement (*Asarnow et al., 2019*; *Zivanov et al., 2018*). Pentavalent and hexavalent sub-particles were extracted using ISECC, our custom implementation of the localized reconstruction technique (*Abrishami et al., 2020*; *Ilca et al., 2015*). ISECC was written for compatibility with RELION 3.1, with additional features for correlative sub-particle analysis (code available at https://github.com/goetschius/isecc/). Sub-particles were locally refined in RELION.

### Model building

All models were built into the corresponding sub-particle maps, rather than icosahedral maps. VP1 models were initialized using an existing structure (PDB: 1sie) after mutating residues to match the A2 strain (*Stehle and Harrison, 1996*). 8A7H5 Fab was initialized using a SWISS-MODEL homology model of an unrelated Fab from mouse mAb 14 (PDB: 3gk8), with manual rebuilding of the CDRs (*Hafenstein et al., 2009*; *Waterhouse et al., 2018*). All models were then refined using sequential rounds of manual building in Coot and automated refinement in PHENIX (*Emsley et al., 2010*; *Liebschner et al., 2019*). Models were validated with MolProbity (*Chen et al., 2010*).

### Data and code availability

Maps and models for the pentavalent and hexavalent capsomers are deposited at wwPDB for both MuPyV (PDB 7K24, 7K25; EMDB 22642, 22643) and the MuPyV-Fab complex (PDB 7K22, 7K23, EMDB 22640, 22641). The icosahedral maps are likewise deposited as EMDB-22645 and EMDB-22646 for MuPyV and the MuPyV-Fab complex, respectively. ISECC, our custom sub-particle extraction program, is available on GitHub (https://github.com/goetschius/isecc; *Goetschius, 2020* (copy archived at https://github.com/elifesciences-publications/isecc)).

### Statistical analysis

Statistical analyses were performed using Prism eight software (GraphPad) using a Mann-Whitney $U$ test, multiple t tests with statistical significance determined by the Holm-Sidak method, or ordinary one-way ANOVA with Tukey's multiple comparisons test. P values of $< 0.05$ were considered significant, and all data are shown as mean, with error bars representing SD. Figures contain the data from all repeats and no data points were excluded. Statistical methods were not used to pre-determine sample sizes. The quantifications of hydrocephalus and kidney VP1 IHC were performed in a blinded manner; no blinding was employed for other experiments. All sample sizes, numbers of repeats, and statistical tests are included in the Figure Legends. In all figures, ns = p > 0.05, *p<0.05, **p<0.01, ***p<0.001, ****p<0.0001. All significant differences are labeled.

## Acknowledgements

The authors thank the staff of the Penn State College of Medicine Flow Cytometry Core Facility, N Sheaffer, J Bednarczyk, J Vogel, and J Zhang for assistance with flow cytometry analysis; G Snavely and E Mullady of the Comparative Medicine Histology Core for the preparation of tissue sections; B Garcea for the generous gift of the PyVP1-pGEX-4T-2 plasmid and rabbit VP1 antisera; and the staff

of the Department of Comparative Medicine at the Penn State College of Medicine. Research reported in this publication was supported by the National Institute of Neurological Disorders and Stroke, the National Institute of Allergy and Infectious Diseases, and the National Cancer Institute under award numbers R01NS088367 and R01NS092662 (AEL), R01AI107121 (SLH), F32NS106730 (CSNW), F31NS083336 (ELF), and T32 CA060395 (MDL and SMB), and the Jake Gittlen Laboratories for Cancer Research (NDC). The content is solely the responsibility of the authors and does not necessarily represent the official views of the National Institutes of Health. In addition, support for this work was provided by Pennsylvania Department of Health CURE funds (SLH).

## Additional information

### Funding

| Funder | Grant reference number | Author |
| --- | --- | --- |
| National Institute of Neurological Disorders and Stroke | R01NS088367 | Aron E Lukacher |
| National Institute of Neurological Disorders and Stroke | R01NS092662 | Aron E Lukacher |
| National Institute of Allergy and Infectious Diseases | R01AI107121 | Susan L Hafenstein |
| National Institute of Neurological Disorders and Stroke | F32NS106730 | Colleen S Netherby-Winslow |
| National Institute of Neurological Disorders and Stroke | F31NS083336 | Elizabeth L Frost |
| National Cancer Institute | T32CA060395 | Matthew D Lauver |
| National Cancer Institute | T32CA060395 | Stephanie M Bywaters |

The funders had no role in study design, data collection and interpretation, or the decision to submit the work for publication.

### Author contributions

Matthew D Lauver, Conceptualization, Data curation, Formal analysis, Validation, Investigation, Visualization, Methodology, Writing - original draft, Writing - review and editing; Daniel J Goetschius, Conceptualization, Data curation, Software, Formal analysis, Validation, Investigation, Visualization, Methodology, Writing - original draft, Writing - review and editing; Colleen S Netherby-Winslow, Katelyn N Ayers, Investigation, Methodology; Ge Jin, Daniel G Haas, Elizabeth L Frost, Sung Hyun Cho, Carol M Bator, Stephanie M Bywaters, Neil D Christensen, Investigation; Susan L Hafenstein, Conceptualization, Formal analysis, Supervision, Funding acquisition, Investigation, Methodology, Writing - original draft, Project administration, Writing - review and editing; Aron E Lukacher, Conceptualization, Formal analysis, Supervision, Funding acquisition, Writing - original draft, Project administration, Writing - review and editing

### Author ORCIDs

Matthew D Lauver https://orcid.org/0000-0002-7001-9730
Daniel J Goetschius https://orcid.org/0000-0002-6052-7141
Katelyn N Ayers https://orcid.org/0000-0001-6156-8685
Aron E Lukacher https://orcid.org/0000-0002-7969-2841

### Ethics

Animal experimentation: All experiments involving mice were conducted with the approval of Institutional Animal Care and Use Committee (Protocol 47619) of The Pennsylvania State University College of Medicine in accordance with the Guide for the Care and Use of Laboratory Animals of the National Institutes of Health. The Pennsylvania State University College of Medicine Animal Resource Program is accredited by the Association for Assessment and Accreditation of Laboratory Animal Care International (AAALAC). The Pennsylvania State University College of Medicine has an Animal

Welfare Assurance on file with the National Institutes of Health's Office of Laboratory Animal Welfare; the Assurance Number is A3045-01.

## Decision letter and Author response

Decision letter https://doi.org/10.7554/eLife.61056.sa1
Author response https://doi.org/10.7554/eLife.61056.sa2

# Additional files

## Supplementary files

• Supplementary file 1. Collection statistics. Independent datasets were collected for MuPyV and the MuPyV-Fab complex.

• Supplementary file 2. Refinement statistics. Local refinement allowed models to be built into the higher resolution capsomer maps.

• Supplementary file 3. Fab contact residues. The majority of Fab contacts are through the heavy chain, with minor contributions from the light chain.

• Supplementary file 4. Statistics of existing and new cryo-EM polyomavirus maps. Capsomer-based local refinement allowed rapid refinement of polyomavirus to high resolution, using only a modest particle number.

• Supplementary file 5. ISECC source code. Archive of the ISECC source code used in sub-particle generation. The most current version of the software is maintained at https://github.com/goetschius/isecc.

• Supplementary file 6. Oligonucleotide sequences. Sequences of oligonucleotides used for site-directed mutagenesis, qPCR, sequencing.

• Transparent reporting form

## Data availability

All maps and models are deposited at wwPDB and their accession numbers are provided in the Data and code availability section of our manuscript. Maps and coordinates (4 zip files) generated during this study are included in the manuscript and supporting files. Source data files have been provided for Figures 4 and 5 and Figure 4—figure supplement 4, and are available on GitHub with the URL provided in the Data and code availability section.

The following datasets were generated:

| Author(s) | Year | Dataset title | Dataset URL | Database and Identifier |
|---|---|---|---|---|
| Goetschius DJ, Hafenstein SL | 2020 | Murine polyomavirus pentavalent capsomer, subparticle reconstruction | https://www.rcsb.org/structure/7K24 | RCSB Protein Data Bank, 7K24 |
| Goetschius DJ, Hafenstein SL | 2020 | Murine polyomavirus hexavalent capsomer, subparticle reconstruction | https://www.rcsb.org/structure/7K25 | RCSB Protein Data Bank, 7K25 |
| Goetschius DJ, Hafenstein SL | 2020 | Murine polyomavirus pentavalent capsomer with 8A7H5 Fab, subparticle reconstruction | https://www.rcsb.org/structure/7K22 | RCSB Protein Data Bank, 7K22 |
| Goetschius DJ, Hafenstein SL | 2020 | Murine polyomavirus hexavalent capsomer with 8A7H5 Fab, subparticle reconstruction | https://www.rcsb.org/structure/7K23 | RCSB Protein Data Bank, 7K23 |
| Goetschius DJ, Hafenstein SL | 2020 | Murine polyomavirus with 8A7H5 Fab (icosahedral reconstruction) | https://www.ebi.ac.uk/pdbe/entry/emdb/EMD-22646 | Electron Microscopy Data Bank, 22646 |
| Goetschius DJ, Hafenstein SL | 2020 | Murine polyomavirus (icosahedral reconstruction) | https://www.ebi.ac.uk/pdbe/entry/emdb/EMD-22645 | Electron Microscopy Data Bank, 22645 |

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
