## [Decision Letter]

**Acceptance summary:**

This study characterizes a mutant murine polyomavirus with a VP1 capsid mutation that is analogous to the mutation in human JC1 virus that causes the serious human disease progressive multifocal leukoencephalopathy (PML). The mutation confers virus escape from immune detection, leading to productive infection and pathogenesis in the mouse CNS. The authors further use cryoEM to demonstrate the structural basis by which the mutant VP1 escapes antibody-mediated neutralization, thus revealing fundamental new insight into viral mutants that can escape immune surveillance and promote disease outcomes.

**Decision letter after peer review:**

Thank you for submitting your article "Antibody Escape by Polyomavirus Capsid Mutation Facilitates Neurovirulence" for consideration by *eLife*. Your article has been reviewed by three peer reviewers, and the evaluation has been overseen by a Reviewing Editor and Anna Akhmanova as the Senior Editor. The following individuals involved in review of your submission have agreed to reveal their identity: Juha Huiskonen (Reviewer #2); Andrew Macdonald (Reviewer #3).

The reviewers have discussed the reviews with one another and the Reviewing Editor has drafted this decision to help you prepare a revised submission.

Summary:

This study characterizes a mutant murine polyomavirus with a VP1 capsid mutation that is analogous to the mutation in human JC1 viruses that causes the serious human disease progressive multifocal leukoencephalopathy (PML). The mutation confers virus escape from immune detection, leading to productive infection and pathogenesis in the mouse CNS. The authors further use cryoEM to demonstrate the structural basis by which the mutant VP1 escapes antibody-mediated neutralization. This study thus combines in vivo pathogenesis with structural biology to reveal fundamental new insight into viral mutants that can escape immune surveillance and promote disease outcomes.

Essential revisions:

Reviewer #1:

Introduction: "immunological perturbations”: some examples should be listed for clarity.

Introduction: at some point (discussion?) additional comments on non-coding changes in PML variants should be made since the literature regards these changes as important for brain tropism.

Discussion, second paragraph: this paragraph is densely technical and unclear.

Figure 1A: The lateral ventricles are markedly increased in size suggesting hydrocephalus and blocked CSF flow. Does this defect thus suggest that the infection functionally affects the choroid plexi? Is this defect seen in PML? Can oliodendrocytic lesions/infection be identified in the mouse as in human?

Figure 3F legend: Experiment not well explained.

Reviewer #2:

Introduction, Results and Discussion: "subvolume" – "sub-particle". Subvolume refers to refinement of 3D particles in sub-tomogram averaging. In the approach presented, it's 2D sub-particles that are refined.

Introduction, second paragraph: “the brain” – “brains” (brains.… of patients)

"Twelve capsomers lay on an icosahedral five-fold vertex surrounded by five neighboring capsomers and are referred to as pentavalent capsomers". This wording is grammatically somewhat ambiguous. Consider changing to: "The capsomers that occupy the twelve icosahedral five-fold vertices are referred to as pentavalent capsomers, as each of them is surrounded by five neighboring capsomer" or similar sentence.

"Atomic resolution structures" – "Atomic models" would be more appropriate as these models are atomistic descriptions of the interactions, but they are inferred from structures at much lower resolution than "atomic resolution" (see below).

"Using a novel.… strategy". It would be advisable to either explain what is novel in the custom local refinement approach. Later in the text it is mentioned that it is a custom implementation of the localized reconstruction approach (Ilca et al., 2015) so it may be advisable to remove the novelty claim (please see also Abrishami et al., 2020).

"We attained atomic resolution and could build" – "We attained sufficient resolution to build". Resolution of 3.1 Å is not "atomic resolution". This is a term that refers resolution of 1 Å and better. "Near-atomic resolution" is often used for this resolution range but even this is an ill-defined term.

Introduction, last paragraph: "revealed" suggested?

Subsection “Cryo EM reconstruction of MuPyV identifies mechanism of VP1 antibody escape by V296F”, second paragraph: There are indeed six Fab densities per asymmetric unit in the icosahedrally averaged map. But the authors should also consider cases where not all VP1s bind the Fab in all symmetry related positions in all of the particles.

"remaining five hexavalent sites". These sites themselves are not hexavalent. Consider changing to "remaining five sites on the hexavalent capsomer".

Subsection “Cryo EM reconstruction of MuPyV identifies mechanism of VP1 antibody escape by V296F”, last paragraph: The claim regarding fully occupancy is too strong and unnecessary. As there are no clashes between the 360 Fabs in the capsid, a capsid could in theory bind up to 360 Fabs. But it seems the authors haven't tested the question of variable occupancy by 3D classification of the sub-particles. Please reword this or add the 3D classification analysis focused on each of the 6 sites in the asymmetric unit.

"By extension, our data support the concept that the VP1 antibody response selects JCPyV variants capable of causing CNS injury." What is meant by "select"? This is confusing as a major neutralizing antibody cannot bind MuPyV-VP1-296F, which is analogous to JCPyV-VP1- S268F, a mutant found in PML causing viruses. So it seems the opposite may be true, the antibody response may miss instead of selecting at least one PML causing mutant (although this hasn't been directly tested).

Subsection “Data and code availability”: The Gitlab link is still behind a password

Supplementary Figures:

Figure 4—figure supplement 3: Spell out "ROYGBV"

Add FSC curves to support resolution estimates.

Reviewer #3:

1) Whilst the data linking the generation of JC escape mutants in response to neutralising anti-VP1 antibodies is robust, it is less clear to me why these mutants show a defined tropism for CNS over kidney. Further, i do not think that this observation is particularly developed in the manuscript. Does the V296F mutant no longer interact with a key receptor(s) on kidney cells that is/are still present on cells of the CNS? Binding assays undertaken in the manuscript show that the mutant is able to bind fibroblasts. Did the authors consider testing proximal tubular epithelial cells ? I am not asking for more experiments, as i consider that the study is sufficient as is, but i would urge the authors to at least discuss the tropism in more detail in the manuscript to provide the reader with a more rounded understanding of the topic.

---

## [Author Response]

Essential revisions:Reviewer #1:Introduction: "immunological perturbations”: some examples should be listed for clarity.

Examples of the immunological deficiencies/perturbations that are associated with PML development have been added (Introduction, second paragraph).

Introduction: at some point (Discussion?) additional comments on non-coding changes in PML variants should be made since the literature regards these changes as important for brain tropism.

We agree and have now emphasized the importance of the NCCR rearrangements for brain tropism and cited a review detailing the current understanding of these rearrangements (Introduction, second paragraph).

Discussion, second paragraph: this paragraph is densely technical and unclear.

We added several clarifications to this paragraph and removed some jargon; however, it is unavoidably technically dense for understanding details of the virus structure solution.

Figure 1A: The lateral ventricles are markedly increased in size suggesting hydrocephalus and blocked CSF flow. Does this defect thus suggest that the infection functionally affects the choroid plexi? Is this defect seen in PML? Can oliodendrocytic lesions/infection be identified in the mouse as in human?

Both viruses induced comparable bilateral dilatation of the lateral ventricles (Figure 2A, 2B). Ventricular enlargement has been reported late in the course of PML and in a case of JCPyV meningitis (Ray et al., 2015, Agnihotri et al., 2014). Loss of periventricular tissue due to infection/inflammation could lead to a compensatory ventricular enlargement rather than obstructive hydrocephalus. Alternatively, disruption of the choroid plexus by infection could result in transudation of plasma into the ventricles or increased CSF production. We have included these possibilities in the Discussion (eleventh paragraph).

MuPyV LT mRNA is detected in FACS-sorted oligodendrocytes from acutely infected mice but at significantly lower levels than in other glia and brain-infiltrating monocytes. VP1 expression, indicating productive infection, is found in astrocytes but not oligodendrocytes (Shwetank et al., 2019, doi: 10.3389/fimmu.2019.00783). This finding is in agreement with work by Kondo et al. showing preferential JCPyV replication in astrocytes over oligodendrocytes in human glial chimeric mice (Kondo et al., 2014). Because we have previously published this data, we would prefer not to discuss it in this manuscript.

Figure 3F legend: Experiment not well explained.

More explanation has been added to the figure legend and the text has been reworded to clarify how the experiment was performed (subsection “V296F confers resistance to a neutralizing VP1 antibody”).

Reviewer #2:Introduction, Results and Discussion: "subvolume" – "sub-particle". Subvolume refers to refinement of 3D particles in sub-tomogram averaging. In the approach presented, it's 2D sub-particles that are refined.

This modification has been made throughout.

Introduction, second paragraph: “the brain” – “brains” (brains.… of patients)

This has been corrected.

"Twelve capsomers lay on an icosahedral five-fold vertex surrounded by five neighboring capsomers and are referred to as pentavalent capsomers". This wording is grammatically somewhat ambiguous. Consider changing to: "The capsomers that occupy the twelve icosahedral five-fold vertices are referred to as pentavalent capsomers, as each of them is surrounded by five neighboring capsomer" or similar sentence.

Thank you for this rewording. This change has been made (Introduction, third paragraph).

"Atomic resolution structures" – "Atomic models" would be more appropriate as these models are atomistic descriptions of the interactions, but they are inferred from structures at much lower resolution than "atomic resolution" (see below).

Corrected in text.

"Using a novel.… strategy". It would be advisable to either explain what is novel in the custom local refinement approach. Later in the text it is mentioned that it is a custom implementation of the localized reconstruction approach (Ilca et al., 2015) so it may be advisable to remove the novelty claim (please see also Abrishami et al., 2020).

Thank you for this suggestion that we have now used in the manuscript. We have also included the Abrishami citation (subsection “Cryo EM reconstruction of MuPyV identifies mechanism of VP1 antibody escape by V296F”, third paragraph).

"We attained atomic resolution and could build" – "We attained sufficient resolution to build". Resolution of 3.1 Å is not "atomic resolution". This is a term that refers resolution of 1 Å and better. "Near-atomic resolution" is often used for this resolution range but even this is an ill-defined term.

This correction has been made in the Introduction.

Introduction, last paragraph: "revealed" suggested?

Text has been modified in the Introduction.

Subsection “Cryo EM reconstruction of MuPyV identifies mechanism of VP1 antibody escape by V296F”, second paragraph: There are indeed six Fab densities per asymmetric unit in the icosahedrally averaged map. But the authors should also consider cases where not all VP1s bind the Fab in all symmetry related positions in all of the particles.

Subsection “Cryo EM reconstruction of MuPyV identifies mechanism of VP1 antibody escape by V296F“, text was modified for clarity to address this issue.

"remaining five hexavalent sites". These sites themselves are not hexavalent. Consider changing to "remaining five sites on the hexavalent capsomer".

This change was made in the subsection “Cryo EM reconstruction of MuPyV identifies mechanism of VP1 antibody escape by V296F”.

Subsection “Cryo EM reconstruction of MuPyV identifies mechanism of VP1 antibody escape by V296F”, last paragraph: The claim regarding fully occupancy is too strong and unnecessary. As there are no clashes between the 360 Fabs in the capsid, a capsid could in theory bind up to 360 Fabs. But it seems the authors haven't tested the question of variable occupancy by 3D classification of the sub-particles. Please reword this or add the 3D classification analysis focused on each of the 6 sites in the asymmetric unit.

This passage has been modified as above (Subsection “Cryo EM reconstruction of MuPyV identifies mechanism of VP1 antibody escape by V296F“).

"By extension, our data support the concept that the VP1 antibody response selects JCPyV variants capable of causing CNS injury." What is meant by "select"? This is confusing as a major neutralizing antibody cannot bind MuPyV-VP1-296F, which is analogous to JCPyV-VP1- S268F, a mutant found in PML causing viruses. So it seems the opposite may be true, the antibody response may miss instead of selecting at least one PML causing mutant (although this hasn't been directly tested).

We thank the reviewer for this suggestion and have rewritten this sentence (Discussion, first paragraph). JCPyV exists as a quasispecies in PML patients (Van Loy et al., 2015). Our data supports the concept that a VP1 mutant virus could evade an antibody response the controls the WT virus, leading to an outgrowth of the mutant virus.

Subsection “Data and code availability”: The Gitlab link is still behind a password

Github transfer has been completed and the new URL added to the subsection “Data and code availability”.

Supplementary Figures:Figure 4—figure supplement 3: Spell out "ROYGBV"

This modification has been made.

Add FSC curves to support resolution estimates.

These FSC curves have been added as Figure 4—figure supplement 2 and are referred to in the subsection “Cryo EM reconstruction of MuPyV identifies mechanism of VP1 antibody escape by V296F”.

Reviewer #3:1) Whilst the data linking the generation of JC escape mutants in response to neutralising anti-VP1 antibodies is robust, it is less clear to me why these mutants show a defined tropism for CNS over kidney. Further, i do not think that this observation is particularly developed in the manuscript. Does the V296F mutant no longer interact with a key receptor(s) on kidney cells that is/are still present on cells of the CNS? Binding assays undertaken in the manuscript show that the mutant is able to bind fibroblasts. Did the authors consider testing proximal tubular epithelial cells ? I am not asking for more experiments, as i consider that the study is sufficient as is, but i would urge the authors to at least discuss the tropism in more detail in the manuscript to provide the reader with a more rounded understanding of the topic.

This is an important question. We see two possibilities that could account for this difference in brain and kidney tropism by VP1 mutant and WT viruses. First, the kidney may selectively express a decoy (aka, pseudo) receptor that diverts virion uptake toward a nonproductive infection pathway. A second possibility is that certain VP1 mutations may abrogate binding to a host cell receptor in the kidney that the WT virus can still engage, whereas the brain expresses a different receptor to which both WT and mutant viruses can bind. We have included these possibilities in the Discussion (eighth paragraph).